# PHDME: PHYSICS-INFORMED DIFFUSION MODELS WITHOUT EXPLICIT GOVERNING EQUATIONS

## ABSTRACT

Diffusion models are expressive priors for generating and predicting data from high-dimensional dynamical systems. Yet, purely data-driven approaches often lack reliability and trustworthiness, motivating growing interest in physics-informed machine learning (PIML). Most existing PIML methods, however, assume access to exact governing equations during training—an assumption that fails when the dynamics are unknown or too complex to model accurately. To address this gap, we introduce PHDME[1] (Port-Hamiltonian Diffusion Model), a physics-informed diffusion framework that learns system dynamics without requiring exact equations. Our approach first trains a Gaussian process distributed Port-Hamiltonian system (GP-dPHS) on limited observations to capture an energy-based representation of the dynamics. The GP-dPHS is then used to generate a physically consistent and diverse dataset for diffusion training. To enforce physics-consistency, we embed the GP-dPHS structure directly into the diffusion training objective through a loss that penalizes deviations from the learned Hamiltonian dynamics, weighted by the GP's predictive uncertainty. After training, we employ conformal prediction to provide distribution-free uncertainty quantification of the generated trajectories. In this way, PHDME is designed for regimes with scarce data and unknown equations, enabling data-efficient, physically valid trajectory generation with calibrated uncertainty estimates.

## 1 INTRODUCTION

Predicting the evolution of complex dynamical systems is central to policy design (Bevacqua et al., 2023), collision avoidance (Missura & Bennewitz, 2019), and long-horizon planning (Li et al., 2025). However, accurate forecasts remain a significant challenge where dynamics involve high nonlinearity and dimensionality, as well as when observational data are sparse and limited. A common constraint in robotics, for instance, where fully instrumenting a soft-bodied manipulator with tactile sensors is often expensive and physically difficult. Furthermore, many of these systems are described by partial differential equations (PDEs), but traditional numerical solvers are computationally expensive, which requires fine-grained spatiotemporal discretization that is overwhelming for real-time control or long-horizon forecasting. To tackle these challenges, various deep learning frameworks have been proposed to learn the underlying dynamics from collected data. Methods like neural ODE (Chen et al., 2018) and neural PDE (Zubov et al., 2021) formulations impose substantial computational cost. Training requires repeated forward time integrations together with backward sensitivity computations through stiff multiscale solvers. The computational cost scales up with prediction horizon, state dimension, and solver stiffness, leading to high runtime and memory usage that force compromises on model fidelity and spatial resolution of the grid. Although alternative frameworks, such as discrete-time autoregressive models, circumvent the integration cost, they introduce challenges of error accumulation over rollouts.

Diffusion models (Sohl-Dickstein et al., 2015) offer a flexible generative prior for forecasting in dynamical systems. Denoising diffusion defines a forward Markov corruption with Gaussian perturbations and trains a reverse process (Ho et al., 2020; Karras et al., 2022) that estimates the score of the data distribution, achieving state-of-the-art synthesis in images (Xia et al., 2023; Xu et al., 2023),

---

[1]Code available at: `https://github.com/InvincibleTdog/PHDME_anonymous`

videos (Ho et al., 2022; Liang et al., 2024), and audio (Guo et al., 2024). In scientific machine learning the key advantage is the ability to represent full predictive distributions rather than single trajectories, which supports inverse problems (Chung et al., 2023) and planning (Römer et al., 2025) under uncertainty. In case of spatiotemporal problem that are encoded as an image 1 or video, the output of the diffusion model is the solution of the PDE over spatial and temporal domain. Nevertheless, standard diffusion models are purely data-driven, so samples may align with dataset statistics while violating the physics that govern the real world. The absence of explicit physics limits performance and reliability and weakens guarantees in applications like safety-critical systems (Tan et al., 2023).

Physics-informed training addresses this gap by constraining learning with governing equations. Classic work like physics-informed Neural Networks (PINNs) (Raissi et al., 2019) ensures that the learning outcomes follow the physics, and recent work has begun to embed such constraints into generative modeling (Shu et al., 2023; Bastek et al., 2024). These approaches typically require that the governing equations are known (except for some unknown parameters) and can be enforced during training. However, in many real systems, the exact governing equations are unknown or prohibitively complex to model, and observations are limited, e.g., modeling the equations of motion of soft robots via first principles is quite challenging due to the highly nonlinear and unstructured dynamics. Under these conditions, standard physics-informed pipelines are difficult to deploy.

**Contribution:** We aim to offer rapid, physically reliable, multi-step dynamic forecasting. In this paper, we propose PHDME, which is built on a Gaussian-process distributed Port-Hamiltonian System (Tan et al., 2024). The Port-Hamiltonian framework provides an expressive yet physically consistent representation for hard-to-model, unstructured dynamics. We learn the governing equations directly from limited observations by fitting a GP-dPHS that models the underlying Hamiltonian of the system. The learned GP-dPHS is then integrated into the diffusion training objective as a physics-consistency term that aligns the score network with Hamiltonian-consistent dynamics across noise levels. This coupling of energy-based representation learning with diffusion training enables data-efficient forecasting that respects physical structure even when governing equations are unavailable. Moreover, the probabilistic deep prior encapsulates a class of partial differential equations dynamics, enabling it to directly generate the PDE solution reliably even under unseen initial conditions, bypassing the need for iterative, numerical PDE solvers.

Our contributions can be summarized as:

- Leveraging a single draw from the diffusion model, PHDME provides fast forecasts for PDE systems where the governing equations are unknown but highly nonlinear. PHDME produces reliable results even when data availability is strictly limited.

- The proposed PHDME uses structured energy representations of the system to make the learning process physically informed. By using the Bayesian nature of the GP, diffusion model training has been weighted by the uncertainties from the data observation stage, which makes it possible to inform and constrain the system with physics without knowing the exact underlying functions.

- We also introduce a conformal prediction as postprocessing of the PHDME, where we not only provide a physically-valid sample given the initial condition, but also provide the uncertainty quantification of the sample. These features make the method suitable for safety-critical applications.

## 2 PRELIMINARY

In this section, we give a brief overview of denoise diffusion models, Gaussian process distributed Port-Hamiltonian systems, and conformal prediction (CP).

### 2.1 DENOISING DIFFUSION MODELS

Diffusion models have demonstrated excellent potential in various domains (Ho et al., 2020; Song & Ermon, 2019; Dhariwal & Nichol, 2021). While recent efforts extend them to time series forecasting (Rasul et al., 2021), super resolution for dynamic prediction (Rühling Cachay et al., 2023), and time-invariant physics-informed generation (Bastek et al., 2024). The spatiotemporal forecasting

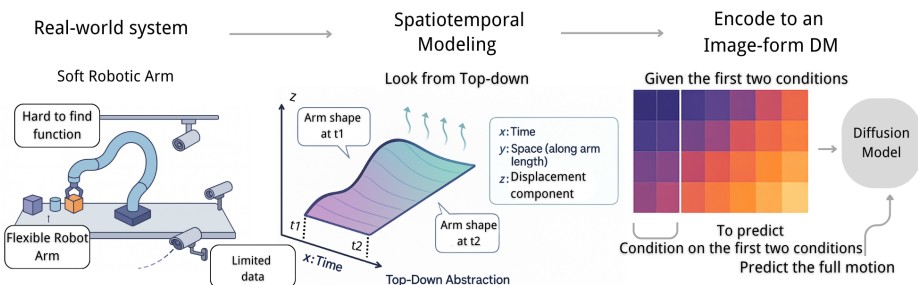

Figure 1: The left panel depicts a typical soft robot scenario in which a flexible continuum manipulator exhibits dynamics that are difficult to specify. The middle panel adopts a top-down parameterization with the y-axis as spatial projection along the arm direction, the z-axis (pixel value) as displacement, and the x-axis as temporal evolution. This converts the evolution into an image form, enabling the diffusion model to synthesize the full spatiotemporal field in a single shot rather than step-by-step rollouts.

with physics guarantee has remained underexplored, especially when the governing equations are unknown or difficult to obtain.

**Diffusion indexing, parameterization, and objective.** Let $\boldsymbol{A}^{(m)}$ be the noised image at step $m \in \{0, \ldots, M\}$, where the diffusion step $m$ is different to any physical time notation $t$. The forward noise corruption is linear Gaussian: $\boldsymbol{A}^{(m)} = \alpha_m \boldsymbol{A}^{(0)} + \sigma_m \boldsymbol{\varepsilon}$ with $\boldsymbol{\varepsilon} \sim \mathcal{N}(\boldsymbol{0}, \mathbf{I})$; the schedule $\{(\alpha_m, \sigma_m)\}_{m=0}^{M}$ yields $\boldsymbol{A}^{(M)} \approx \mathcal{N}(\boldsymbol{0}, \mathbf{I})$. In the $x_0$ parameterization, a neural denoiser predicts the clean sample from a noised input and optional condition $\mathbf{y}$ via $\widehat{\boldsymbol{A}}^{(0)} = f_\theta([\boldsymbol{A}^{(m)}, \mathbf{y}], m)$. The reverse transition uses the closed-form Gaussian posterior with mean $\boldsymbol{\mu}_m(\boldsymbol{A}^{(m)}, \widehat{\boldsymbol{A}}^{(0)})$ and variance $\tilde{\sigma}_m^2 \mathbf{I}$, both fixed by the forward schedule. And $w_m$ is set to Min-SNR-5 weighting (Hang et al., 2023). Training minimizes a timestep-weighted reconstruction loss

$$\mathcal{L}_{\text{DDPM}}(\theta) = \mathbb{E}_{m, \boldsymbol{A}^{(0)}, \boldsymbol{\varepsilon}}\left[w_m \|f_\theta([\alpha_m \boldsymbol{A}^{(0)} + \sigma_m \boldsymbol{\varepsilon}, \mathbf{y}], m) - \boldsymbol{A}^{(0)}\|^2\right].$$

**Sampling and uncertainty.** Starting from $\boldsymbol{A}^{(M)} \sim \mathcal{N}(\boldsymbol{0}, \mathbf{I})$, generation iterates $\boldsymbol{A}^{(m-1)} = \boldsymbol{\mu}_m(\boldsymbol{A}^{(m)}, \widehat{\boldsymbol{A}}^{(0)}) + \tilde{\sigma}_m \boldsymbol{z}$ with $\boldsymbol{z} \sim \mathcal{N}(\boldsymbol{0}, \mathbf{I})$; repeated runs thus form an ensemble approximating the conditional distribution of $\boldsymbol{A}^{(0)}$ given $\mathbf{y}$.

## 2.2 Gaussian Process Distributed Port-Hamiltonian System

Based on Hamiltonian dynamics, GP-dPHS is a physics-informed PDE learning method that not only generalizes well from sparse data, but also provides uncertainty quantification (Tan et al., 2024). The composition of Hamiltonian systems through input and output ports leads to Port Hamiltonian systems, a class of dynamical systems in which ports formalize interactions among components. The Hamiltonian can also be interpreted as the energy representation of the system. This framework applies in the classical finite dimensional setting (Beckers et al., 2022) and extends naturally to distributed parameter and multivariable cases. In the infinite dimensional formulation, the interconnection, damping, and input and output matrices are replaced by matrix differential operators that do not explicitly depend on the state or energy variables. Under this learning structure, once the Hamiltonian is specified, the system model follows in a systematic manner. This general formulation is versatile enough to represent various PDEs and has been shown to capture a wide range of physical phenomena, including heat conduction, piezoelectricity, and elasticity. In what follows, we recall the definition of distributed Port Hamiltonian systems as presented in (Macchelli et al., 2004).

More formally, let $\mathcal{Z}$ be a compact subset of $\mathbb{R}^n$ representing the spatial domain, and consider a skew-adjoint constant differential operator $J$ along with a constant differential operator $G_d$. Define the Hamiltonian functional $\mathcal{H}: \mathcal{X} \to \mathbb{R}$ in this following form:

$$\mathcal{H}(\boldsymbol{x}) = \int_{\mathcal{Z}} H(z, x) dV,$$

where $H\colon \times \mathcal{X} \to \mathbb{R}$ is the energy density. Denote by $\mathcal{W}$ the space of vector-valued smooth functions on $\partial\mathcal{Z}$ representing the boundary terms $\mathcal{W} := \{w | w = B_\mathcal{Z}(\delta_{\boldsymbol{x}}\mathcal{H}, \boldsymbol{u})\}$ defined by the boundary operator $B_\mathcal{Z}$. Then, the general formulation of a multivariable dPHS $\Sigma$ is fully described by

$$\Sigma(J, R, \mathcal{H}, G) = \begin{cases} \frac{\partial \boldsymbol{x}}{\partial t} = (J - R)\delta_{\boldsymbol{x}}\mathcal{H} + G_d \boldsymbol{u} \\ \boldsymbol{y} = G_d^* \delta_{\boldsymbol{x}}\mathcal{H} \\ w = B_\mathcal{Z}(\delta_{\boldsymbol{x}}\mathcal{H}, \boldsymbol{u}), \end{cases} \tag{1}$$

where $R$ is a constant differential operator taking into account energy dissipation. Furthermore, $\boldsymbol{x}(t, \boldsymbol{z}) \in \mathbb{R}^n$ denotes the state (also called energy variable) at time $t \in \mathbb{R}_{\geq 0}$ and location $\boldsymbol{z} \in \mathcal{Z}$ and $\boldsymbol{u}, \boldsymbol{y} \in \mathbb{R}^m$ the I/O ports, see (Tan et al., 2024) for more details. Generally, the $J$ matrix defines the interconnection of the elements in the dPHS, whereas the Hamiltonian $H$ characterizes their dynamical behavior. The constitution of the $J$ matrix predominantly involves partial differential operators. The port variables $\boldsymbol{u}$ and $\boldsymbol{y}$ are conjugate variables in the sense that their duality product defines the energy flows exchanged with the environment of the system.

When the system dynamics are only partially known, the Hamiltonian can be modeled within a probabilistic framework using Gaussian processes. A Gaussian process is fully specified by a mean function and a covariance function, and as a nonparametric Bayesian prior, it is well-suited for smooth Hamiltonian functionals. Its invariance under linear transformations further supports consistent representation propagation through the operators that define the dynamics (Jidling et al., 2017).

Integrating these concepts, the unknown Hamiltonian latent function of a distributed system is encoded within a dPHS model to ensure physical consistency. Here, the unknown dynamics are captured by approximating the Hamiltonian functional with a GP, while treating the matrices $J$, $R$, and $G$ (more precisely, their estimates $\hat{J}_\Theta$, $\hat{R}_\Theta$, and $\hat{G}_\Theta$) as hyperparameters. This leads to the following GP representation for the system dynamics:

$$\frac{\partial \boldsymbol{x}}{\partial t} \sim \mathcal{GP}(\hat{G}_\Theta \boldsymbol{u}, k_{dphs}(\boldsymbol{x}, \boldsymbol{x}')),$$

with a physics-informed kernel function defined as

$$k_{dphs}(\boldsymbol{x}, \boldsymbol{x}') = \sigma_f^2 (\hat{J}\hat{R}_\Theta)\delta_{\boldsymbol{x}} \exp\left(-\frac{\|\boldsymbol{x} - \boldsymbol{x}'\|^2}{2\varphi_l^2}\right)\delta_{\boldsymbol{x}'}^\top (\hat{J}\hat{R}_\Theta)^\top,$$

where $\hat{J}\hat{R}_\Theta = \hat{J}_\Theta - \hat{R}_\Theta$ and the kernel is based on the squared exponential function. The training of this GP-dPHS model involves optimizing the hyperparameters $\Theta$, $\varphi_l$, and $\sigma_f$ by minimizing the negative log marginal likelihood. Hence, the physics representation prior is learned by GP without any presumption of the functional form; this information is fully described by the structured mean function and variance function.

Exploiting the linear invariance property of GPs, the Hamiltonian $\hat{\mathcal{H}}$ now follows a GP prior. This integration effectively combines the structured, physically consistent representation of distributed Port-Hamiltonian systems with the flexibility of GP to handle uncertainties and learn unknown dynamics from data. The resulting framework not only ensures that the model adheres to the underlying physics but also provides a comprehensive, data-informed prediction of the system's behavior.

## 2.3 CONFORMAL PREDICTION

Conformal prediction a statistical technique used to quantify the uncertainty of predictions in machine learning models. It provides a prediction set that contains the true output with a user-specified probability $1 - \delta$. We calibrate the mean squared error of our stochastic generator with conformal prediction. Let the calibration set be $\mathcal{D}_{\text{cal}} = \{\mathbf{x}_i^\star\}_{i=1}^K$, where each $\mathbf{x}_i^\star$ is the ground-truth dynamic landscape on the grid $\mathcal{G}$. For every $i$ we call the predictor $Num$ times, drawing $\widehat{\mathbf{x}}_i^{(n)} \sim \mathsf{P}_\theta(\cdot)$, $n = 1, \ldots, N$, and define the non-conformity score (NCS) of a single sample as (Lindemann et al., 2023; Vlahakis et al., 2024):

$$r_{i,n} = \frac{1}{|\mathcal{G}|}\left\|\widehat{\mathbf{x}}_i^{(n)} - \mathbf{x}_i^\star\right\|_F^2. \tag{2}$$

Pooling the $K * Num$ scores and sorting them in ascending order gives an empirical error distribution for a single stochastic draw. For a target miscoverage level $\delta \in (0, 1)$, set the calibrated threshold to the order statistic

$$\tau := \text{Quantile}_{(1 + \frac{1}{K * Num})(1 - \delta)}(r^{(1)}, \ldots, r^{(K * Num)}), \tag{3}$$

Under exchangeability of scores, we can say under at least $1 - \delta$ probability guarantee, a future prediction from $\mathsf{P}_\theta(\cdot)$ has mean squared error at most $\tau_\delta$, formally as: $\mathbb{PROB}(r \leq \tau) \geq 1 - \delta$.

## 3 PROPOSED PHDME

In this section, we will discuss the assumptions and problem formulation, followed by a detailed introduction of the proposed **P**ort-**H**amiltonian **D**iffusion **M**odel without **E**xplicit underlying equations (PHDME) enhances predictive performance by leveraging the learned energy representations and observation uncertainties.

### 3.1 ASSUMPTIONS AND SETTINGS

We study the problem of spatiotemporal dynamic prediction with uncertainty quantification. We have a PDE system $0 = f(\mathbf{x}, \mathbf{dx}, \cdots)$ and aim to predict the solution for $t = 0, \cdots, T$ for this system. We assume that this system can be written in dPHS form even though we do not require knowledge about the components. further we assume that we can collect limited data from the PDE.

Hence, instead of learning the regular dynamic directly, where the underlying functions are hard to acquire. We transform the problem space to the structured derivative space. The energy representation can be modeled through a distributed Port-Hamiltonian system. We adopt a dPHS representation in which the dynamics are modeled as

$$\frac{\partial \boldsymbol{x}}{\partial t} = (J - R)\delta_{\boldsymbol{x}}\hat{\mathcal{H}} + G_d \boldsymbol{u}$$

where $J$ is power preserving, $R$ is dissipative, $G_d$ maps inputs, and $\hat{\mathcal{H}}$ is a learned Hamiltonian functional. In PHDME, $\hat{\mathcal{H}}$ is represented by a Gaussian process trained on limited observations, and the induced Hamiltonian gradients are integrated into the diffusion training objective through a physics consistency term. This aligns the learned score field with Hamiltonian consistent dynamics across noise levels and avoids reliance on guidance during sampling.

We make the following assumptions:

**Assumption 1** *The PHDME is designed to handle the scenario where the observations are limited and the underlying functions are hard to acquire, which means the regular data-driven predictors are hard to train and the conventional physics-informed methods are not able to handle. We observe the state on a limited spatiotemporal grid, yielding measurements $\left\{\boldsymbol{x}(t_i, z_j)\right\}_{i=1,\ldots,N_t}^{j=1,\ldots,N_z}$.*

**Assumption 2** *The structural form of the interconnection, dissipation, and input operators is known up to a finite set of parameters. Specifically, $J$, $R$, and $G_d$ are specified by templates with unknown coefficients $\Theta \subset \mathbb{R}^{n_\Theta}$, which are estimated from data. The qualitative structure, such as the type of friction model encoded in $R$, is known, while the numerical values of the parameters may be unknown.*

### 3.2 PHDME FRAMEWORK

Instead of forecasting by sequential rollouts or numerical integration, which can be computationally expensive, PHDME generates the entire future spatiotemporal field in a single pass conditioned on the given initial conditions. The central idea is to guide this single draw-image like generation with a deep prior learned from limited observations. The training pipeline has two stages, as illustrated in Figure 2. First, we encode the scarce observations from the real system through the dPHS structure. And naturally learn a probabilistic energy-based representation of the system using the Gaussian processes. Then this deep prior is used to synthesize a rich dataset for the diffusion model training, as well as guiding the second training stage of the PHDME with a physics consistency loss derived from the prior and weighted by its predictive uncertainty based on observations, thereby aligning the learned score field with Hamiltonian consistent dynamics while preserving data efficiency.

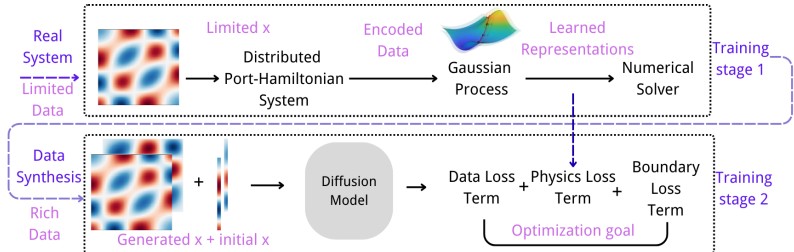

Figure 2: This figure visualize the two-stage training of the PHDME, where we firstly train a rather slow but structured deep prior. Then we leverage this prior to inform the diffusion training for rapid sample generations.

**Data collection and GP-dPHS training (stage 1).** We observe the state $\mathbf{x}(t, z)$ at discrete times and spatial locations, $\mathcal{D} = \left\{ t_i, \; z_j, \; \mathbf{x}(t_i, z_j), \; \boldsymbol{u}(t_i) \right\}_{i=0, j=0}^{i=N_t-1, j=N_z-1}$. Since measurements are sparse and derivatives are required for model learning, we fit a smooth Gaussian process interpolant to $x(t, z)$ using a squared exponential kernel, and exploit closed form differentiation of the Gaussian process to collect over time yields $X = \left[ \tilde{\boldsymbol{x}}(t_0), \ldots, \tilde{\boldsymbol{x}}(t_{N_t-1}) \right], \dot{X} = \left[ \partial_t \tilde{\boldsymbol{x}}(t_0), \ldots, \partial_t \tilde{\boldsymbol{x}}(t_{N_t-1}) \right]$, and the training set $\mathcal{E} = \left[ X, \dot{X} \right]$, aligned with the input sequence $\{\boldsymbol{u}(t_i)\}_{i=1}^{N_t}$. This construction provides derivative information from $x(t, z)$ while enlarging spatial coverage for subsequent GP-dPHS training. Based on this dataset, we learn a distributed Port Hamiltonian representation in which the dynamics satisfy

$$\partial_t \boldsymbol{x}(t, z) = (J - R)\, \delta_{\boldsymbol{x}} \hat{\mathcal{H}}(t, z) + G_d\, \boldsymbol{u}(t, z),$$

with interconnection matrix $J$, dissipative term $R$, and Hamiltonian $\hat{\mathcal{H}}$. The unknown Hamiltonian is modeled by a Gaussian process and eventually unknown coefficients of $J$, $R$, and $G_d$ are treated as hyperparameters $\Theta$. Using the linear invariance of Gaussian processes, we place a GP prior on the energy derivatives and obtain a GP over the time derivative of the state,

$$\partial_t \boldsymbol{x} \sim \mathcal{GP}\big(\hat{G}_\Theta\, \boldsymbol{u}, \; k_{dphs}(\boldsymbol{x}, \boldsymbol{x}')\big),$$

with physics-informed kernel

$$k_{dphs}(\boldsymbol{x}, \boldsymbol{x}') = \sigma_f^2\, (\hat{J}_\Theta - \hat{R}_\Theta)\, \delta_{\boldsymbol{x}}\, \exp\Big(-\frac{\|\boldsymbol{x}-\boldsymbol{x}'\|^2}{2\varphi_l^2}\Big)\, \delta_{\boldsymbol{x}'}^\top\, (\hat{J}_\Theta - \hat{R}_\Theta)^\top.$$

We train the model on $\mathcal{E}$ by maximizing the marginal likelihood with respect to $\Theta$ and the kernel hyperparameters $(\varphi_l, \sigma_f)$. The resulting posterior induces a stochastic Hamiltonian $\hat{\mathcal{H}}$ and yields the learned dPHS

$$\partial_t \boldsymbol{x}(t, z) = (\hat{J}_\Theta - \hat{R}_\Theta)\, \delta_{\boldsymbol{x}} \hat{\mathcal{H}}(t, z) + \hat{G}_\Theta\, \boldsymbol{u}(t, z),$$

which serves as a probabilistic physics prior for subsequent data generation and diffusion training. However, since this numerical solution of GP-dPHS is computational demanding, we train a physics-informed diffusion model instead of directly using the GP-dPHS for prediction.

**Dataset generation using GP samples.** We place a GP prior over the energy functional, yielding a posterior that captures a family of plausible energy representations. Using only the posterior mean to represent the learned dynamics neglects posterior uncertainty and is therefore not a valid surrogate for the true system. Instead, we leverage random fourier feature prior draw to provide realizations of the GP-dPHS that will be used as training data for the diffusion model. See appendix A.5 for more details.

GP-dPHS yields a posterior over Hamiltonian energy functionals rather than a single estimate based on limited observations, as discussed in 3.2. We draw function realizations of the Hamiltonian gradient $\delta_{\boldsymbol{x}} \hat{\mathcal{H}}$ from this posterior and then plug it into the dPHS form and solve it numerically to generate a trajectory $\mathbf{x}(z, t)$. For a real and shift-invariant kernel $k_f$, Bochner theory (Langlands, 2006) implies a spectral density that admits a finite feature approximation. We use a random feature map:

$$\phi(\mathbf{x}) = \sqrt{\tfrac{2}{d}}\, [\cos(\boldsymbol{\omega}_1^\top \mathbf{x} + \beta_1), \ldots, \cos(\boldsymbol{\omega}_D^\top \mathbf{x} + \beta_D)]^\top,$$

with $\boldsymbol{\omega}_j$ drawn from the spectral density and $\beta_j \sim \mathrm{Uniform}[0, 2\pi]$. The $d$ denotes the dimensions of the feature map, so that $k_f(\mathbf{x}, \mathbf{x}') \approx \phi(\mathbf{x})^\top \phi(\mathbf{x}')$. A pathwise prior sample is

$$f(\mathbf{x}) = \phi(\mathbf{x})^\top \mathbf{w}, \qquad \mathbf{w} \sim \mathcal{N}(\mathbf{0}, \mathbf{I}),$$

which provides one realization for the stacked gradients $\delta_{\boldsymbol{x}}\mathcal{H}$. Then the posterior correction on a finite set can be occupied. For any query set $X = \big[\mathbf{x}^\star(t_0), \ldots, \mathbf{x}^\star(t_{N_t-1})\big]$, define the covariance kernel: $\mathbf{C}_{XX} = \big[k_f(\mathbf{x}_i, \mathbf{x}_j)\big]_{ij}$, $\mathbf{C}_{\star X} = \big[k_f(\mathbf{x}_i^\star, \mathbf{x}_j)\big]_{ij}$. A posterior function sample on $X_\star$ is then

$$f(\mathbf{x}^\star) = \mu_f(\mathbf{x}^\star) + f(\mathbf{x}^\star) + \mathbf{C}_{\star X}\big(\mathbf{C}_{XX} + \sigma_n^2 \mathbf{I}\big)^{-1}\big(\mathbf{y} - \mu_f(X) - f(X)\big),$$

applied component wise to $\delta_{\boldsymbol{x}}\mathcal{H}$. This warps the prior draw to match the observations and yields an exact finite dimensional posterior sample suitable for insertion into the dPHS evolution. By evaluating the sample-based posterior function under different initial conditions, we build a rich training and validation set for the PHDME.

**Diffusion training Notation.** We follow the diffusion indexing introduced above. The latent at diffusion step $m \in \{0, \ldots, M\}$ is $\boldsymbol{A}^{(m)}$, the clean tensor is $\boldsymbol{A}^{(0)}$, and conditioning is provided by the first two frames $\mathbf{c}_{\mathrm{init}}$. The denoiser $f_\theta$ predicts the clean tensor in the $\boldsymbol{A}_0$ parameterization,

$$\widehat{\boldsymbol{A}}^{(0)} = f_\theta\big([\boldsymbol{A}^{(m)}, \mathbf{c}_{\mathrm{init}}], m\big).$$

We interpret $\widehat{\boldsymbol{A}}^{(0)}$ as the image like representation of the state over the spatiotemporal grid $\mathcal{T} \times \mathcal{Z}$, where $\mathcal{T} = \big\{ t_0, \cdots, t_i \big\}_{i=0}^{i=N_t-1}$. and $\mathcal{Z} = \big\{ z_0, \cdots, z_j \big\}_{j=0}^{j=N_z-1}$. aligned with $\mathbf{c}_{\mathrm{init}}$.

**Physics operator from GP-dPHS.** Purely data-driven models can fit trajectories while ignoring invariants or stability. Following the spirit of Bastek et al. (2024), we therefore regularize the denoiser with a physics operator. In contrast to approaches that assume a closed-form PDE, we first learn a deep probabilistic physics prior with the GP-dPHS model and then use this learned Hamiltonian representation to define the residual. Concretely, on the discrete spatio-temporal grid we interpret the denoised sample $\widehat{\boldsymbol{A}}^{(0)}$ together with the conditioning $\mathbf{c}_{\mathrm{init}}$ as a candidate field $\boldsymbol{x}$. The dPHS residual operator $\mathcal{F}_{\mathrm{dPHS}}(\boldsymbol{x})$ is obtained by evaluating the port-Hamiltonian dynamics from equations 1 with the GP-based energy gradients and discretizing the spatial and temporal derivatives by centered differences in the interior and consistent boundary stencils. We then aggregate this residual into a scalar penalty with boundary terms

$$\mathcal{R}_{\mathrm{phys}}\big(\widehat{\boldsymbol{A}}^{(0)}; \mathbf{c}_{\mathrm{init}}\big) = \frac{1}{|\Omega|}\big\|\mathcal{F}_{\mathrm{dPHS}}(\boldsymbol{x})\big\|_2^2 + \lambda_{\mathrm{bc}}\,\mathcal{B}\big(\boldsymbol{x}; \mathbf{c}_{\mathrm{init}}\big), \tag{4}$$

where $|\Omega|$ denotes the total number of grid points (space, time, and batch) and $\mathcal{B}$ encodes fixed-end and conditioning constraints (see Appendix A.7 for details).

Because GP-dPHS is Bayesian, it provides a posterior variance at each grid location that quantifies the epistemic uncertainty in the learned Hamiltonian vector field. We exploit this by constructing an uncertainty-aware version $\widetilde{\mathcal{R}}_{\mathrm{phys}}$, in which the contribution of each residual term is weighted by the inverse GP variance. In regions where the learned physics representation is confident, the physics penalty is strong; in regions with high uncertainty it is softened. This turns the physics operator into a heteroscedastic regularizer that guides the diffusion model toward Hamiltonian-consistent trajectories where the prior is reliable, without over-constraining it where the model is less certain.

**Proposed PHDME loss.** PHDME augments the standard reconstruction objective with the physics penalty evaluated on the denoised prediction,

$$\mathcal{L}(\theta) = \mathop{\mathbb{E}}_{m, \boldsymbol{A}^{(0)}, \boldsymbol{\varepsilon}} \Big[w_m \big\|f_\theta\big([\boldsymbol{A}^{(m)}, \mathbf{c}_{\mathrm{init}}], m\big) - \boldsymbol{A}^{(0)}\big\|_2^2 + \lambda_{\mathrm{phys}}\,\widetilde{\mathcal{R}}_{\mathrm{phys}}\big(f_\theta([\boldsymbol{A}^{(m)}, \mathbf{c}_{\mathrm{init}}], m); \mathbf{c}_{\mathrm{init}}\big)\Big].$$

The second term backpropagates through $f_\theta$ and aligns the learned score field with Hamiltonian consistent dynamics across diffusion steps. It does not alter the forward noising process or the ancestral form of the reverse kernel.

**Generative uncertainty via conformal prediction.** Beyond producing physically plausible trajectories, we would like PHDME to expose *calibrated* generative uncertainty for its forecasts. To this end, we use CP as a purely post-hoc layer on top of the trained diffusion model; the training objective of PHDME is unchanged. For a fixed conditioning $\mathbf{c}_{\mathrm{init}}$, each reverse-diffusion rollout of DDPM with an independent noise seed can be viewed as one sample from the learned conditional distribution $p_\theta(\mathbf{x} \mid \mathbf{c}_{\mathrm{init}})$. These samples are exchangeable by construction, which is exactly the assumption under which CP provides finite-sample coverage guarantees.

Concretely, we construct a calibration set $\mathcal{D}_{\mathrm{cal}} = \{(\mathbf{c}_{\mathrm{init}}^{(i)}, \mathbf{x}_i^\star)\}_{i=1}^K$ and, for each $\mathbf{c}_{\mathrm{init}}^{(i)}$, draw $Num$ stochastic PHDME rollouts to compute the non-conformity scores defined in equation 2. The empirical $(1 - \delta)$-quantile of these scores yields a threshold $\tau_\delta$ as in equation 3, so that for a new conditioning and an on-the-fly test rollout $r$ we have $\mathbb{P}(r \leq \tau_\delta) \geq 1 - \delta$. In other words, the CP layer wraps PHDME's stochastic samples into prediction bands with guaranteed marginal coverage under the data-generating distribution. Appendix A.8 reports the resulting coverage and set sizes in detail, empirically confirming that our CP construction behaves as expected in this setting.

## 4 EXPERIMENTS

In this section, we demonstrate the effectiveness and performance of the PHDME. By quantitatively discuss the accuracy and generative speed, and qualitatively visualizing the generated sample, we show the powerful aspects of the framework. The further discussion can be found in the appendix.

### 4.1 SETUP

**Data benchmarks.** PHDME is designed to have access only to a small number of trajectories, not to closed-form PDEs. To mimic this regime, we evaluate the proposed method on three PDE systems: (i) a canonical fixed-end string governed by the wave equation, see Appendix A.1 (ii) a one-dimensional shallow-water system, see Appendix A.2 and (iii) a **real-world vibrating spring recorded by a high frame-rate camera** in Appendix A.3

For the two simulator-based benchmarks (string and shallow water), the high-fidelity solvers are used only to generate a small observation set of 20 trajectories on the $64 \times 64$ grid, with an additional temporal downsampling factor of 50 to reflect limited sensing and logging capabilities. We fit a GP-dPHS model to these sparse observations and then use GP-dPHS sampling (Section 3.2) for data augmentation to produce 10 000 training and 1 000 validation trajectories per system. A separate set of 10 000 trajectories from the simulators serves as ground-truth test data for evaluating accuracy. For the real-world benchmark, we use a high-speed Blackfly S USB3 camera (Flir BFS-U3-16S2C-CS) to record the motion of a red spring at 226 FPS. The spring body is segmented using an RGB mask, yielding a binary foreground mask. We then skeletonize this mask to obtain a centerline representation of the flexible spring (see Figure 3). We learn a GP-dPHS prior to these processed trajectories and use GP-dPHS sampling to generate 4,500 training and 500 validation trajectories for PHDME training; the remaining real-world data from the camera are held out as a test set to assess transfer to truly equation-unknown data. More details of data preprocessing are in the Appendix A.3.

For all experiments, PHDME operates on the $64 \times 64$ grid using a U-Net (Ronneberger et al., 2015) backbone whose input and output dimensions match this grid. We adopt a 4-channel design: two channels encode the initial frames $\mathbf{c}_{\mathrm{init}}$, and two channels collect the dynamic fields $(p, q)$. The deep priors are trained from limited observations; PHDME never sees or uses the analytic PDE form, only the learned deep prior is used to inform the physics.

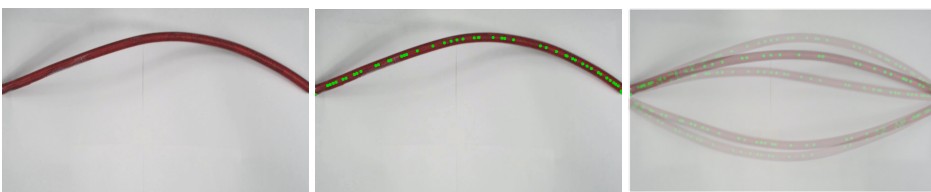

Figure 3: From left to right, the figure illustrates the process starting with the original video, skeletonization, and final spring movement data over time (overlay).

**Baselines.** We compare PHDME against four baselines under the same architecture, training schedule, and hardware: (i) a standard diffusion model with the same U-Net and diffusion schedule but no physics loss; (ii) a diffusion model with limited physics that only encodes easy-to-observe structure (e.g., fixed-end boundary conditions) without access to deep Hamiltonian priors; (iii) a GP-dPHS integrator that rolls out trajectories step-by-step using learned energy-based representations; and (iv) a NeuralODE (Chen et al., 2018) that learns purely from data without any knowledge of the underlying physics (and thus cannot use physics-based data augmentation). Baselines (i)–(ii) test whether PHDME's learned physics prior improves over purely data-driven or weakly constrained generative modeling, while (iii) isolates the benefit of a strong GP-dPHS prior and (iv) provides a fully data-driven baseline reference.

## 4.2 RESULTS

**Quantitative results** We present the grid-average metrics for the PDEs in Table 1, the models with physics knowledge (Boundary condition / GP-dPHS priors) starts with lower Loss during the early stage of training, then balance the data loss and physics loss terms as the training iterations increase.

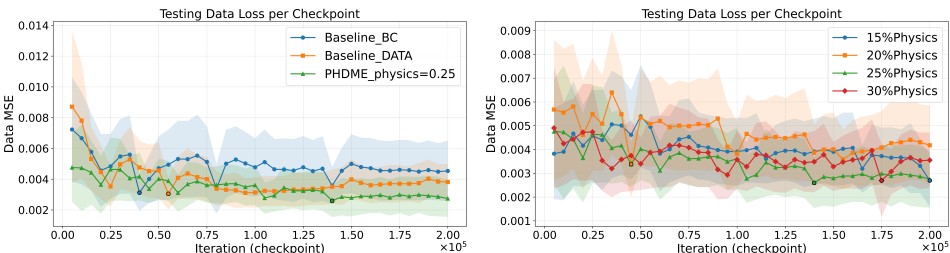

Figure 4: On the left side, PHDME beats the baselines with pure-data driven and limited physics access by having the minimum MSE over iterations. On the right side, we further investigate the potential impacts of the physics-loss term percentage regarding the performance.

Table 1 summarizes test performance across all datasets. We emphasize that the CP module is designed for the diffusion generative uncertainty; hence, the non-conformity score (NCS) (recall in 2) is only reported for diffusion-based models. On the synthetic string and 1-D shallow water benchmark, PHDME attains the lowest MSE, reducing error by roughly $28\%$ relative to the standard DDPM, while also achieving the smallest NCS. Indicating more accurate and better calibrated generative uncertainty than both purely data-driven and weakly physics-aware diffusion baselines. On the real-world spring dataset, PHDME matches the best purely data-driven DDPM baseline in MSE (within about $4\%$) and remains competitive in NCS. In all three settings, the baselines like GP-dPHS and NeuralODE exhibit substantially larger MSE, illustrating the difficulty of long-horizon rollouts in the sparse state space grid and highlighting that amortizing the GP-dPHS priors into a diffusion model leads to more accurate generative predictions. Notice that the data augmentation step is usually on a dense grid to have accurate derivatives, but for fair comparison against the diffusion models, we list the result of GP-dPHS on the same $64 \times 64$ grid here.

| Dataset | Metric | **Models** | | | | |
|---|---|---|---|---|---|---|
| | | PHDME | DDPM | DDPM+Limited Physics | GP-dPHS | NeuralODE |
| String | MSE | **2.74**e$-$**3** | 3.81e$-$3 | 4.53e$-$3 | 2.03e$-$2 | 2.54e$-$2 |
| Dataset | NCS | **6.41**e$-$**3** | 6.82e$-$3 | 9.80e$-$3 | $-$ | $-$ |
| Shallow | MSE | **2.06**e$-$**2** | 2.31e$-$2 | 2.92e$-$2 | 2.23e$-$1 | 4.76e$-$2 |
| Water | NCS | **9.75**e$-$**2** | 1.02e$-$1 | 1.05e$-$1 | $-$ | $-$ |
| Real-world | MSE | 5.21e$-$2 | **5.02**e$-$**2** | 5.44e$-$2 | 7.65e$-$1 | 2.037 |
| Spring | NCS | 2.93e$-$3 | 3.07e$-$3 | **7.09**e$-$**4** | $-$ | $-$ |

Table 1: The results of model test performance. Key take-away: The data augmentation from GP-dPHS improve the general performance, and the deep physics priors help the model to learn the dynamics pattern better with general tighter generative uncertainty bound.

**Qualitative results**  We qualitatively assess the full PHDME pipeline by inspecting predicted states. Our success criterion is accurate state forecasting under unseen initial conditions and environments. As illustrated in Fig. 5, generated samples closely match the true system behavior, preserving boundary behavior and phase progression. See (App. A.6) that errors introduced by the GP-dPHS data generation and by diffusion sampling are both limited on the evaluation grid. And the sampling process has been visualized in Fig. 6. For more details and discussions regarding the representation learning and the correctness of learned Hamiltonian, see App. A.7.

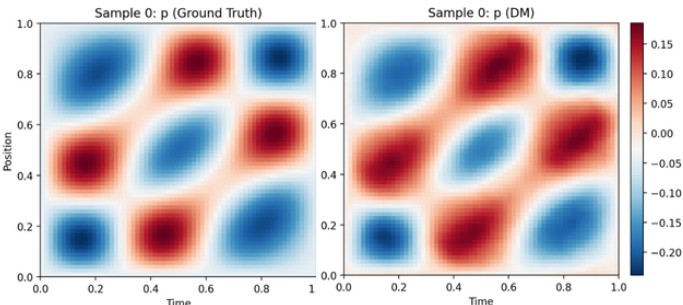

Figure 5: Left: Ground-truth state evolution of the wave equation. Right: Physically consistent and accurate prediction of PHDME based on sparse data and limited knowledge of the governing equations. Key takeaway: PHDME generates samples with correct dynamic pattern and amplitude by only conditioning on the initial two frames.

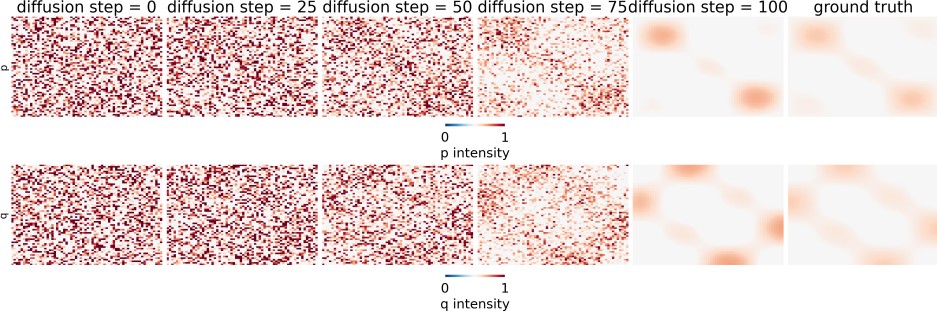

Figure 6: The visualization of diffusion sampling process

**Related work.**  We target the realistic regime where only limited trajectory data are available and the exact governing functions are inaccessible, as formalized in Assumption 1. In this setting, equation-based diffusion models (Jacobsen et al., 2025; Bastek et al., 2024) that require an explicit PDE are not directly applicable, since we restrict all methods to observed trajectories and simple, easily measured constraints (e.g., boundary conditions). At the other extreme, purely data-driven continuous-time models such as NeuralODEs (Chen et al., 2018) lack structural physics priors and perform poorly under these limited observations. For more discussions on NeuralODE, see App. A.9. PHDME instead learns a reusable physics prior from data and amortizes it into a diffusion model that operates directly on trajectories without ever accessing explicit equations.

## 5    CONCLUSION

We presented PHDME, a physics-informed diffusion framework for dynamical systems in the realistic regime where only limited trajectories are observed and the exact governing equations are unavailable. By learning a reusable GP-dPHS prior and amortizing it into a denoising diffusion model, PHDME combines the flexibility of diffusion models with structure-aware representations of Hamiltonian dynamics. Across synthetic string, 1-D shallow water, and real-world spring systems, PHDME framework provide fairly reliable predictions. Our results highlight representation level physics priors as a promising guide for generative modeling of dynamical systems under scarce physical supervision.

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

# A  APPENDIX

## A.1  WAVE PDE DATA GENERATION

We synthesize supervision using a physically faithful simulator of a fixed–end string that solves the one–dimensional wave equation on a fine grid and then projects to the learning grid. Let $s(z,t)$ denote displacement, with spatial domain $z \in [0, L]$ and time $t \in [0, T]$. The continuous dynamics satisfy

$$\partial_{tt}s(z,t) = c^2 \, \partial_{zz}s(z,t), \qquad s(0,t) = 0, \; s(L,t) = 0, \tag{5}$$

with initial conditions $s(z,0) = s_0(z)$ and $\partial_t s(z,0) = w_0(z)$. The learned state is the derivative pair

$$p(z,t) = \partial_z s(z,t), \qquad q(z,t) = \partial_t s(z,t), \tag{6}$$

and we collect the state vector as $\mathbf{x}(z,t) = [p(z,t), \, q(z,t)]$.

**Fine–to–coarse simulation.**  We integrate an equivalent first–order system on a fine grid and then downsample to the learning resolution. Let $y(t) = [s(\cdot, t); \, w(\cdot, t)]$ with $w = \partial_t s$. Discretize space on $N_z^{\text{fine}}$ nodes with step $\Delta z^{\text{fine}}$, and approximate the Laplacian by a second–order central stencil. The semi–discrete dynamics are

$$\frac{\mathrm{d}}{\mathrm{d}t} \begin{bmatrix} s \\ w \end{bmatrix} = \begin{bmatrix} w \\ c^2 \, \mathbf{D}_{\text{fine}}^{zz} \, s \end{bmatrix}, \qquad s_1(t) = s_{N_z^{\text{fine}}}(t) = 0, \tag{7}$$

where $\mathbf{D}_{\text{fine}}^{zz}$ is the tridiagonal second–difference operator with Dirichlet boundary rows. We integrate (7) over $N_t^{\text{fine}}$ fine time points using an adaptive ODE solver. From the fine solution we compute

$$p^{\text{fine}} = \partial_z s \approx \mathbf{D}_{\text{fine}}^z \, s, \qquad q^{\text{fine}} = \partial_t s \approx \mathbf{D}_{\text{fine}}^t \, s, \tag{8}$$

with $\mathbf{D}_{\text{fine}}^z$ the centered first–difference in $z$ and $\mathbf{D}_{\text{fine}}^t$ a centered time stencil. Optional Gaussian smoothing with standard deviation $\sigma$ may be applied to $s$ before differencing. We then downsample $(p^{\text{fine}}, q^{\text{fine}})$ to the learning grid of size $N_z \times N_t$ to obtain

$$p \in \mathbb{R}^{N_z \times N_t}, \qquad q \in \mathbb{R}^{N_z \times N_t}. \tag{9}$$

**Randomized, valid initial conditions.**  To span smooth, physically consistent excitations, we sample $s_0$ and $w_0$ as finite Fourier–sine series that respect fixed ends:

$$s_0(z) = \sum_{n=1}^{N_{\mathrm{m}}} a_n \sin\left(\frac{n\pi z}{L}\right) \cos \phi_n, \qquad w_0(z) = \sum_{n=1}^{N_{\mathrm{m}}} a_n \sin\left(\frac{n\pi z}{L}\right) \sin \phi_n \, \frac{n\pi c}{L}, \tag{10}$$

with amplitudes $a_n$ in a symmetric range and phases $\phi_n \sim \mathcal{U}[0, 2\pi]$.

**Four–channel tensor with boundary conditioning.**  Each sample is packaged into

$$\big[ \texttt{p\_field}, \, \texttt{full\_p}, \, \texttt{q\_field}, \, \texttt{full\_q} \big] \in \mathbb{R}^{4 \times N_z \times N_t},$$

where $\texttt{full\_p} = p$ and $\texttt{full\_q} = q$. The conditioning channels encode the first two time frames with zeros elsewhere. The first frame of $\texttt{p\_field}$ is set to zero to anchor the spatial–slope channel:

$$\texttt{p\_field}[:,0] = \mathbf{0}, \quad \texttt{p\_field}[:,1] = p[:,1], \quad \texttt{p\_field}[:,t] = \mathbf{0} \text{ for } t \geq 2, \tag{11}$$

$$\texttt{q\_field}[:,0] = q[:,0], \quad \texttt{q\_field}[:,1] = q[:,1], \quad \texttt{q\_field}[:,t] = \mathbf{0} \text{ for } t \geq 2. \tag{12}$$

**Normalization.**  To harmonize dynamic range, we apply channelwise min–max normalization to $[\ell, u]$ with $\ell = -1$ and $u = 1$,

$$\widetilde{X} = \ell + \frac{u - \ell}{X_{\max} - X_{\min} + \epsilon} \big( X - X_{\min} \big), \qquad X \in \{\texttt{full\_p}, \texttt{full\_q}\}, \tag{13}$$

and use the same affine map for the corresponding conditioning frames.

---

**Algorithm 1** Wave Data Generation (create_string_dataset v3.0, $z$–space, $s$–displacement)

---

1: Set $N_z^{\text{fine}} \leftarrow 4N_z$ and $N_t^{\text{fine}} \leftarrow 4N_t$
2: **for** each sample **do**
3:    Sample $\{a_n, \phi_n\}_{n=1}^{N_m}$ and construct $s_0, w_0$ via (10)
4:    Integrate (7) on the fine grid to obtain $s^{\text{fine}}(z_i, t_j)$
5:    Compute $p^{\text{fine}} = \partial_z s^{\text{fine}}$ and $q^{\text{fine}} = \partial_t s^{\text{fine}}$ using centered differences
6:    Downsample $p^{\text{fine}}, q^{\text{fine}}$ to $p, q \in \mathbb{R}^{N_z \times N_t}$
7:    Set targets full_p $\leftarrow p$ and full_q $\leftarrow q$
8:    Form conditioning p_field, q_field with the first two frames and zeros elsewhere, enforcing p_field$[:, 0] = \mathbf{0}$
9:    Apply channelwise normalization and write tensors to disk
10: **end for**

---

### A.2 SHALLOW WATER PDE DATA GENERATION

We synthesize an additional supervision set from a one–dimensional linearized shallow–water layer, again using a high–resolution PDE solver followed by projection to the learning grid. Let $\eta(x, t)$ denote the free–surface displacement above rest, with spatial domain $x \in [0, L]$ and time $t \in [0, T]$. For each sample we draw a water depth $H$ uniformly from a range $[H_{\min}, H_{\max}]$ and set the wave speed

$$c(H) = \sqrt{g\,H},$$

with gravitational constant $g$. The continuous dynamics follow the linearized shallow–water equation

$$\partial_{tt}\eta(x,t) = c(H)^2\,\partial_{xx}\eta(x,t), \qquad \eta(0,t) = 0,\ \eta(L,t) = 0, \tag{14}$$

with initial height $\eta(x, 0) = \eta_0(x)$ and initial vertical velocity $\partial_t\eta(x, 0) = v_0(x)$. The learned state is again a pair of spatial–temporal derivatives,

$$p(x,t) = \partial_t\eta(x,t), \qquad q(x,t) = \partial_x\eta(x,t), \tag{15}$$

and we collect $\mathbf{x}(x,t) = [p(x,t),\ q(x,t)]$ as the representation used by PHDME.

**Fine to coarse shallow water simulation.**    We integrate an equivalent first–order system on a fine grid and then downsample to the learning resolution. Define $y(t) = [\eta(\cdot, t);\ v(\cdot, t)]$ with $v = \partial_t\eta$. Discretize space on $N_x^{\text{fine}}$ nodes with step $\Delta x^{\text{fine}}$ and approximate the second derivative with a centered three–point stencil. The semi–discrete dynamics read

$$\frac{\mathrm{d}}{\mathrm{d}t}\begin{bmatrix} \eta \\ v \end{bmatrix} = \begin{bmatrix} v \\ c(H)^2\,\mathbf{D}_{\text{fine}}^{xx}\,\eta \end{bmatrix}, \qquad \eta_1(t) = \eta_{N_x^{\text{fine}}}(t) = 0, \tag{16}$$

where $\mathbf{D}_{\text{fine}}^{xx}$ is the tridiagonal second–difference operator with Dirichlet boundary rows. In code we set $L = T = 1$, choose a learning grid of size $N_x \times N_t = 64 \times 64$, and a fine grid $N_x^{\text{fine}} = 4N_x$, $N_t^{\text{fine}} = 4N_t$ to obtain accurate derivatives. The ODE (16) is integrated over $N_t^{\text{fine}}$ fine time points using a high–order adaptive solver (DOP853). Dirichlet boundary conditions are enforced by pinning the endpoint time–derivatives, so that $\eta$ and $v$ at $x = 0, L$ remain fixed over time.

From the fine solution $\eta^{\text{fine}}(x_i, t_j)$ we compute the representation fields

$$p^{\text{fine}} = \partial_t\eta \approx \mathbf{D}_{\text{fine}}^t\,\eta^{\text{fine}}, \qquad q^{\text{fine}} = \partial_x\eta \approx \mathbf{D}_{\text{fine}}^x\,\eta^{\text{fine}}, \tag{17}$$

where $\mathbf{D}_{\text{fine}}^t$ and $\mathbf{D}_{\text{fine}}^x$ are centered finite–difference stencils along $t$ and $x$, respectively. For numerical stability we apply a mild Gaussian smoothing in time to $\eta^{\text{fine}}$ before differencing (with standard deviation scaled to the fine temporal grid). We then downsample $(p^{\text{fine}}, q^{\text{fine}})$ by uniform subsampling in both $x$ and $t$ to the learning grid

$$p, q \in \mathbb{R}^{N_x \times N_t}.$$

To remove residual numerical drift in the temporal derivative at initialization, we enforce $p(\cdot, 0) = 0$.

**Randomized initial conditions and depth.** To span a family of smooth, physically consistent shallow–water excitations, each sample draws a depth $H \sim \mathcal{U}[H_{\min}, H_{\max}]$ and a random finite sine series for the initial free surface,

$$\eta_0(x) = \sum_{n=1}^{N_{\mathrm{m}}} a_n \sin\left(\frac{n\pi x}{L}\right), \qquad \partial_t \eta(x, 0) = v_0(x) \equiv 0, \tag{18}$$

where $N_{\mathrm{m}} \in \{1, \ldots, 5\}$ is sampled uniformly and amplitudes $a_n$ are drawn from a symmetric range $[a_{\min}, a_{\max}]$ with random sign. The sine basis automatically satisfies the fixed–end constraint $\eta_0(0) = \eta_0(L) = 0$, and the zero–velocity initialization is consistent with our choice to set $p(\cdot, 0) = 0$ during data generation.

**Four channel tensor with boundary conditioning.** Each shallow–water trajectory is packaged into a four–channel tensor

$$\left[\texttt{p\_field}, \texttt{full\_p}, \texttt{q\_field}, \texttt{full\_q}\right] \in \mathbb{R}^{4 \times N_x \times N_t},$$

with

$$\texttt{full\_p} = p, \quad \texttt{full\_q} = q,$$

and boundary–conditioning channels that expose the first few time frames and mask out the rest:

$$\texttt{p\_field}[:, t] = \begin{cases} p[:, t], & t = 0, 1, \\ \mathbf{0}, & t \geq 2, \end{cases} \qquad \texttt{q\_field}[:, t] = \begin{cases} q[:, t], & t = 0, 1, \\ \mathbf{0}, & t \geq 2. \end{cases} \tag{19}$$

Thus $\texttt{p\_field}$ and $\texttt{q\_field}$ encode two observed frames of the temporal and spatial derivatives of $\eta$, while $\texttt{full\_p}$ and $\texttt{full\_q}$ provide the full spatiotemporal evolution that the model must reconstruct.

**Global normalization.** To harmonize dynamic range across samples while preserving relative amplitudes, we use global min–max normalization per derivative type. In a first pass over the dataset we collect

$$p_{\min}, p_{\max} = \min_{i,x,t} p^{(i)}(x, t), \ \max_{i,x,t} p^{(i)}(x, t), \qquad q_{\min}, q_{\max} = \min_{i,x,t} q^{(i)}(x, t), \ \max_{i,x,t} q^{(i)}(x, t),$$

where $i$ indexes samples. In a second pass we rescale every occurrence of $p$ and $q$ (both conditioning and target channels) to a fixed range $[\ell, u] = [-1, 1]$ via

$$\widetilde{p} = \ell + \frac{u - \ell}{p_{\max} - p_{\min} + \epsilon} (p - p_{\min}), \qquad \widetilde{q} = \ell + \frac{u - \ell}{q_{\max} - q_{\min} + \epsilon} (q - q_{\min}), \tag{20}$$

yielding normalized tensors $\widetilde{\texttt{p\_field}}, \widetilde{\texttt{full\_p}}, \widetilde{\texttt{q\_field}}, \widetilde{\texttt{full\_q}}$ that are fed to the model.

## A.3 REAL WORLD SPRING DATA COLLECTION

**Experimental setup.** As an abstraction of deformable obstacles that robots may encounter in their operational environments, we consider a Home Depot extension spring (model #26455, length $41.91\,\mathrm{cm}$, diameter $1.42\,\mathrm{cm}$) mounted with fixed endpoints and excited into transverse oscillation. This setup mimics the dynamic behavior of flexible obstacles such as swinging cables or a soft robot arm.

**High speed acquisition and coordinate system.** We record the motion of the spring using a high–speed RGB camera (Blackfly S USB3 Flir BFS-U3-16S2C-CS) at $226\,\mathrm{FPS}$. Let $i \in \{0, \ldots, N_t^{\mathrm{raw}} - 1\}$ index video frames and $t_i = i\,\Delta t$ with $\Delta t = \frac{1}{226}\,\mathrm{s}$ denote the corresponding physical time. The spring endpoints are rigidly mounted and their pixel locations are identified once at the beginning of each recording. Using the known physical length $L = 41.91\,\mathrm{cm}$, we define a normalized arclength coordinate $z \in [0, L]$ along the spring and interpolate the extracted centerline onto a fixed grid $\{z_j\}_{j=1}^{N_z}$, resulting in discrete observations of the transversal deflection $s(t_i, z_j)$.

**Algorithm 2** Shallow–Water Data Generation (create_shallow_water_dataset v3.0, $x$–space, $\eta$–displacement)

---

1: Set $N_x, N_t \leftarrow 64$ and $N_x^{\text{fine}} \leftarrow 4N_x$, $N_t^{\text{fine}} \leftarrow 4N_t$
2: **for** each sample **do**
3:     Sample depth $H \sim \mathcal{U}[H_{\min}, H_{\max}]$ and set $c \leftarrow \sqrt{gH}$
4:     Sample $\{a_n\}_{n=1}^{N_m}$ and build $\eta_0$ via (18) on $N_x$ grid points
5:     Set $v_0(x) \equiv 0$ and form $y(0) = [\eta_0; v_0]$
6:     Integrate (16) on the fine grid to obtain $\eta^{\text{fine}}(x_i, t_j)$
7:     Compute $p^{\text{fine}} = \partial_t \eta^{\text{fine}}$ and $q^{\text{fine}} = \partial_x \eta^{\text{fine}}$ using centered differences with optional Gaussian smoothing
8:     Downsample $p^{\text{fine}}, q^{\text{fine}}$ to $p, q \in \mathbb{R}^{N_x \times N_t}$ and enforce $p(\cdot, 0) = 0$
9:     Set targets full_p $\leftarrow p$, full_q $\leftarrow q$
10:     Form conditioning channels p_field, q_field by copying the first two time frames and setting later frames to zero
11: **end for**
12: Compute global $(p_{\min}, p_{\max})$ and $(q_{\min}, q_{\max})$ over all samples
13: **for** each sample **do**
14:     Apply global min–max normalization to all four channels and write the $4 \times N_x \times N_t$ tensor to disk
15: **end for**

---

**RGB segmentation and skeletonization.** To facilitate robust segmentation, the spring is painted red and the background is chosen to provide strong color contrast. For each frame, we apply a simple RGB thresholding mask

$$[150, 0, 0] \ \leq \ \text{RGB}(x, y) \ \leq \ [255, 80, 80]$$

componentwise to isolate spring pixels from the background. The resulting binary mask encodes the body of the deformable obstacle. We then apply skeletonization to reduce the segmented region to a one–pixel–wide medial axis (Fig. 7), which provides a compact representation of the spring shape. Mapping this skeleton to the arclength grid $\{z_j\}$ produces a discrete, time–indexed centerline

$$\mathcal{D}_0 = \big\{(t_i, z_j, s(t_i, z_j))\big\},$$

where $s(t_i, z_j)$ denotes the transversal deflection at time $t_i$ and arclength position $z_j$ measured in pixel units and later scaled to physical units via the known length $L$.

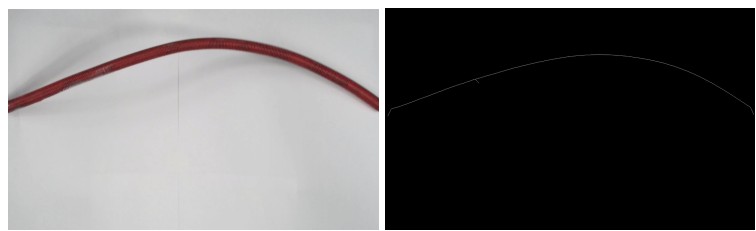

Figure 7: The visualization of the skeletonization process

**Denoising and Gaussian process representation.** Real world videos contain sensor noise, quantization artifacts, and occasional segmentation errors. To mitigate these effects while preserving the underlying dynamics, we first apply a Kalman filter along the temporal axis of each spatial location $z_j$ to obtain a denoised trajectory $\hat{s}(t_i, z_j)$. The denoised dataset

$$\hat{\mathcal{D}}_1 = \big\{(t_i, z_j, \hat{s}(t_i, z_j))\big\}$$

is then used to train a Gaussian Process (GP) model that acts as a continuous, differentiable representation of the spring dynamics. We treat $(t, z) \in [0, T] \times [0, L]$ as the input and model the state

$$s(t, z) \sim \mathcal{GP}\big(m(t, z), \ k\big((t, z), (t', z')\big)\big),$$

with a smooth covariance kernel $k$, e.g. a squared–exponential kernel in time and space, and mean function $m$ fitted from data. Training proceeds by maximizing the GP marginal likelihood on $\hat{\mathcal{D}}_1$, resulting in a posterior distribution $p\big(s(\cdot,\cdot) \mid \hat{\mathcal{D}}_1\big)$.

Because derivatives of a GP are again GPs, the posterior directly provides a probabilistic estimate of temporal and spatial partial derivatives,

$$\frac{\partial s}{\partial t}(t, z), \qquad \frac{\partial s}{\partial z}(t, z), \tag{21}$$

which we use as physics–aware latent features. We define the representation fields

$$p(t, z) = \frac{\partial s}{\partial z}(t, z), \qquad q(t, z) = \frac{\partial s}{\partial t}(t, z), \tag{22}$$

consistent with the synthetic string system, and evaluate $(p, q)$ on a refined spatial grid of size $N_e \gg N_z$ to obtain a dense, noise–robust surrogate of the real–world string dynamics.

### A.4    Synthesize Dataset using Mean Prediction of GP-dPHS

This section describes how the version 4 data generator constructs spatiotemporal training pairs by simulating the mean field dynamics implied by a trained Gaussian–Process distributed Port–Hamiltonian system. The generator replaces the analytical wave operator with the posterior mean of two Gaussian Processes that approximate the Hamiltonian gradients and then integrates the induced first–order evolution to produce full fields of $p$ and $q$.

**Learned energy gradients and induced evolution**    Let $u(x,t)$ denote displacement on $x \in [0, L]$ and $t \in [0, T]$. The representation uses

$$p(x,t) = \partial_t u(x,t), \qquad q(x,t) = \partial_x u(x,t), \tag{23}$$

stacked channelwise into $\boldsymbol{x}(x,t) = [p(x,t),\ q(x,t)]$. The GP dPHS module comprises two Gaussian Processes trained on pairs $(p, q)$ to regress the energy gradients $g_p = \partial E/\partial p$ and $g_q = \partial E/\partial q$. Denote their posterior means by

$$\mu_p(p,q) = \mathbb{E}[g_p(p,q) \mid \mathcal{D}], \qquad \mu_q(p,q) = \mathbb{E}[g_q(p,q) \mid \mathcal{D}], \tag{24}$$

where $\mathcal{D}$ is the training set of derivative–integral pairs. The distributed Port–Hamiltonian evolution induced by these learned gradients is

$$\partial_t p(x,t) = \partial_x \mu_q\big(p(x,t), q(x,t)\big), \qquad \partial_t q(x,t) = \partial_x \mu_p\big(p(x,t), q(x,t)\big), \tag{25}$$

with fixed–end constraints applied at the spatial boundaries for the $p$ channel. Equation (25) specializes the canonical dPHS structure to the GP mean and consequently yields a learned but physically structured flow on the representation.

**Space–time discretization and solver**    Discretize the spatial domain on $S$ nodes with spacing $\Delta x$ and the time horizon on $T$ frames with step $\Delta t$. Let $\mathbf{D}_x$ be the standard centered first–difference matrix on the interior nodes with Dirichlet boundary handling. Vectorize the state at time $t$ as $\mathbf{x}(t) \in \mathbb{R}^{2S}$ with $\mathbf{x}(t) = [\mathbf{p}(t); \mathbf{q}(t)]$. The right–hand side used by the integrator is

$$\frac{\mathrm{d}}{\mathrm{d}t} \begin{bmatrix} \mathbf{p}(t) \\ \mathbf{q}(t) \end{bmatrix} = \begin{bmatrix} \mathbf{D}_x\,\mu_q\big(\mathbf{p}(t), \mathbf{q}(t)\big) \\ \mathbf{D}_x\,\mu_p\big(\mathbf{p}(t), \mathbf{q}(t)\big) \end{bmatrix}, \tag{26}$$

where $\mu_p$ and $\mu_q$ are evaluated pointwise at each spatial node using the trained GP posterior means. A standard explicit adaptive ODE solver advances (26) over $[0, T\Delta t]$. After each step, boundary rows of $\mathbf{p}$ are set to zero to enforce fixed ends, which preserves the intended physical interpretation of $p$ at the string endpoints.

**Initialization and conditioning convention**    The generator samples smooth, band–limited initial profiles that satisfy the boundary conditions. The convention follows the version 3 setup for compatibility with the downstream diffusion model. The first frame of the $p$ channel is set to zero and the first two frames of the $q$ channel are provided by the sampler. The solver then integrates (26) forward in time to obtain a complete trajectory $\{\mathbf{p}(t_j), \mathbf{q}(t_j)\}_{j=0}^{T-1}$ on the learning grid. This seeding strategy anchors the learned representation on early frames and stabilizes the subsequent generative steps.

**Mean only synthesis and uncertainty handling**   The evolution in (26) uses the posterior means $\mu_p, \mu_q$ exclusively to synthesize ground truth. This choice yields a single, coherent physical trajectory per initialization without injecting GP sampling noise, which is desirable when creating supervisory targets for representation learning. The GP predictive variances are retained as optional quality indicators for out–of–distribution detection during generation and can be logged for later analysis but do not perturb the synthesized fields.

**Packaging and normalization**   For each realization the generator writes a four–channel tensor of shape $[4, S, T]$,

$$\big[\,\texttt{p\_field},\ \texttt{full\_p},\ \texttt{q\_field},\ \texttt{full\_q}\,\big]. \tag{27}$$

The targets are $\texttt{full\_p} = \mathbf{p}$ and $\texttt{full\_q} = \mathbf{q}$. The conditioning channels encode the two initial time frames with zeros elsewhere and respect the initialization convention for $p$. A channelwise affine normalization maps the targets to a symmetric range with the same transform applied to the corresponding conditioning frames to maintain consistency.

### A.5   Synthesize Dataset using Sample Prediction of GP-dPHS

**From limited observations to a generative physics prior.**   Let $\mathcal{D} = \{(\mathbf{x}_n, y_n)\}_{n=1}^{N}$ be a small set of observations used to train a Gaussian process distributed Port Hamiltonian system. The Gaussian process does not return a single function, it yields a posterior distribution over Hamiltonian energy functionals. We exploit this posterior to draw function realizations of the energy gradients and to simulate many physically consistent trajectories $\mathbf{x}(t, z) = [\,p(t, z),\ q(t, z)\,]^{\top}$ for diffusion training.

**Random Fourier feature prior draw.**   Consider a real, continuous, shift invariant kernel $k_f(\cdot, \cdot)$ for the gradient field. By Bochner theory there exists a spectral density $\rho(\boldsymbol{\omega})$ such that

$$k_f(\mathbf{x}, \mathbf{x}') = \int_{\mathbb{R}^d} e^{\,\mathrm{i}\,\boldsymbol{\omega}^{\top}(\mathbf{x} - \mathbf{x}')}\,\rho(\boldsymbol{\omega})\,d\boldsymbol{\omega}.$$

We approximate $k_f$ by a random $D$ dimensional feature map

$$\phi(\mathbf{x}) = \sqrt{\tfrac{2}{D}}\,\big[\,\cos(\boldsymbol{\omega}_1^{\top}\mathbf{x} + \beta_1), \ldots, \cos(\boldsymbol{\omega}_D^{\top}\mathbf{x} + \beta_D)\,\big]^{\top},$$

with $\boldsymbol{\omega}_j \sim \rho$ and $\beta_j \sim \mathrm{Uniform}[0, 2\pi]$. This gives $k_f(\mathbf{x}, \mathbf{x}') \approx \phi(\mathbf{x})^{\top}\phi(\mathbf{x}')$. A pathwise prior sample is then

$$f_0(\mathbf{x}) = \phi(\mathbf{x})^{\top}\mathbf{w}, \qquad \mathbf{w} \sim \mathcal{N}(\mathbf{0}, \mathbf{I}),$$

which provides one random realization for the stacked energy gradients $f(\mathbf{x}) = \big[d\hat{E}/dp(\mathbf{x}),\ d\hat{E}/dq(\mathbf{x})\big]^{\top}$.

**Posterior correction on a finite set.**   Let $X = [\mathbf{x}_1, \ldots, \mathbf{x}_N]$ collect the training inputs and let $\mathbf{y}$ collect the targets. Denote the learned mean by $\mu_f(\cdot)$. Define the covariance blocks

$$\mathbf{C}_{XX} = \big[k_f(\mathbf{x}_i, \mathbf{x}_j)\big]_{ij}, \quad \mathbf{C}_{\star X} = \big[k_f(\mathbf{x}_i^{\star}, \mathbf{x}_j)\big]_{ij}, \quad \mathbf{C}_{\star\star} = \big[k_f(\mathbf{x}_i^{\star}, \mathbf{x}_j^{\star})\big]_{ij},$$

for any query set $X_{\star} = \{\mathbf{x}_j^{\star}\}_{j=1}^{N_{\star}}$. The random Fourier feature draw induces the vector $f_0(X)$ and its evaluation on $X_{\star}$, written $f_0(X_{\star})$. A function sample from the posterior on $X_{\star}$ is obtained by the exact conditioning correction

$$f(\mathbf{x}^{\star}) \ = \ \mu_f(\mathbf{x}^{\star}) \ + \ f_0(\mathbf{x}^{\star}) \ + \ \mathbf{C}_{\star X}\big(\mathbf{C}_{XX} + \sigma_n^2\mathbf{I}\big)^{-1}\Big(\mathbf{y} - \mu_f(X) - f_0(X)\Big), \tag{28}$$

applied entrywise to both gradient components. Equation (28) warps the prior draw so that it agrees with the observations in a kernel consistent manner, and it yields an exact posterior sample in the finite dimensional sense induced by $X$ and $X_{\star}$.

**Insertion into the distributed Port Hamiltonian dynamics.** The sampled gradients define the variational derivative $\delta_{\mathbf{x}}\hat{\mathcal{H}}(\cdot) = \left[ d\hat{E}/dp(\cdot),\, d\hat{E}/dq(\cdot) \right]^{\top}$. On the spatial grid we assemble the semi discrete evolution

$$\frac{d}{dt}\begin{bmatrix}\mathbf{p}(t)\\\mathbf{q}(t)\end{bmatrix} = \mathbf{A}\,\delta_{\mathbf{x}}\hat{\mathcal{H}}(\mathbf{p}(t),\mathbf{q}(t)) + \mathbf{B}\,\mathbf{u}(t),$$

where $\mathbf{A}$ is the discrete representation of $J - R$ and boundary conditions, and $\mathbf{B}$ maps inputs. The right hand side is evaluated by applying centered differences in the interior and consistent one sided stencils at the boundaries to the sampled gradient fields. With initial state fixed by the first two frames, we integrate in time with an adaptive Runge Kutta scheme to obtain the trajectories

$$\text{full\_p} = \{\mathbf{p}(t_i)\}_{i=1}^{N_t}, \qquad \text{full\_q} = \{\mathbf{q}(t_i)\}_{i=1}^{N_t}.$$

**Assembly of conditioning and targets.** The conditioning channels keep only the first two frames,

$$\text{p\_field}(:,:,1\text{:}2) = \text{full\_p}(:,:,1\text{:}2), \qquad \text{q\_field}(:,:,1\text{:}2) = \text{full\_q}(:,:,1\text{:}2),$$

and are zero elsewhere. Stacking $[\text{p\_field}, \text{full\_p}, \text{q\_field}, \text{full\_q}]$ yields a tensor of shape $[4, N_z, N_t]$ that matches the diffusion model interface.

**Why this sample based generator helps representation learning.** Drawing $\delta_{\mathbf{x}}\hat{\mathcal{H}}$ from the posterior produces a family of Hamiltonian consistent vector fields that reflect epistemic uncertainty learned from $\mathcal{D}$. The resulting collection of simulated trajectories covers a diverse yet physically structured region of the state space. This enlarged dataset serves as supervision for the diffusion objective, which we further weight by the predictive uncertainty, thereby aligning the learned score field with the Port Hamiltonian manifold while remaining data efficient.

**Implementation notes in v5.0.** The code fixes the trained hyperparameters, constructs the random Fourier feature map, draws $\mathbf{w}$ to obtain $f_0$, and applies the posterior correction in (28) on the grid required by the discrete operator. Each dataset shard records the random seeds, solver tolerances, grid sizes $(N_z, N_t)$, and identifiers of the hyperparameters to ensure exact reproducibility of the sampled gradient fields and of the generated trajectories.

## A.6 DISPLACEMENT RECONSTRUCTION FROM $(p, q)$ AND VALIDATION PROTOCOLS

**State, operators, and learned surrogates.** On a spatial grid $\mathcal{Z}$ and discrete time index $t = 0, \ldots, T$, the port-Hamiltonian state is $(p_t(z), q_t(z))$. The GP-dPHS learns the Hamiltonian gradients as functions on the grid, yielding surrogates $\hat{g}_p(p, q) \approx \partial H/\partial p$ and $\hat{g}_q(p, q) \approx \partial H/\partial q$ (implemented by the two trained heads loaded from `model_dp_trained.pth` and `model_dq_trained.pth`). In the canonical wave-form system, the continuous-time dynamics are $\dot{q} = \partial H/\partial p$ and $\dot{p} = -\partial H/\partial q$; we therefore define $\widehat{dq}(p, q) := \hat{g}_p(p, q)$ and $\widehat{dp}(p, q) := -\hat{g}_q(p, q)$. Boundary handling follows the PDE module used during training (Dirichlet by default in our code), and the time step $\Delta t$ is read from the dataset metadata.

**Displacement reconstruction (rollout).** Given two initial frames $(p_0, q_0)$ and $(p_1, q_1)$ on $\mathcal{Z}$, we reconstruct the entire displacement trajectory $\{q_t\}_{t=2}^{T}$ by iterating an explicit, symplectic first-order update (vectorized over $z \in \mathcal{Z}$):

$$q_{t+1} = q_t + \Delta t\,\widehat{dq}(p_t, q_t), \qquad p_{t+1} = p_t + \Delta t\,\widehat{dp}(p_t, q_t), \qquad t = 1, \ldots, T-1.$$

In practice, we: (i) load the GP-dPHS checkpoints and the dataset item containing *initial* two frames (`p_init`, `q_init`); (ii) standardize/unstardardize using the same statistics as training; (iii) loop the update above for $T-2$ steps; (iv) enforce the boundary condition after each step. The reconstructed displacement is the sequence $\{q_t\}$.

**How this appears in the codebase.** Data are formatted as four channels (`p_full`, `p_init`, `q_full`, `q_init`) by the dataset scripts (`create_string_dataset_v5.py`). The GP models are defined and loaded from `train_gp_phs_v35.py`, while the port-Hamiltonian residuals and utilities reside in `pde.py` and `residuals_string.py`. The diffusion model (`unet_model.py` with sampling utilities in `denoising_utils.py`/`main.py`) consumes the same conditioning (`p_init`, `q_init`) to generate trajectory samples that are evaluated against the ground truth produced by the GP-dPHS simulator.

**Validation protocol.** We validate two aspects: (A) the *physics fidelity* of GP-dPHS rollouts; (B) the *data efficiency and accuracy* of the diffusion model trained on GP-dPHS trajectories.

1. **GP-dPHS accuracy.** For a set of random initializations, compare $\{q_t\}$ reconstructed by the GP-dPHS integrator to the reference simulator (same grid and $\Delta t$). Report MSE scores.

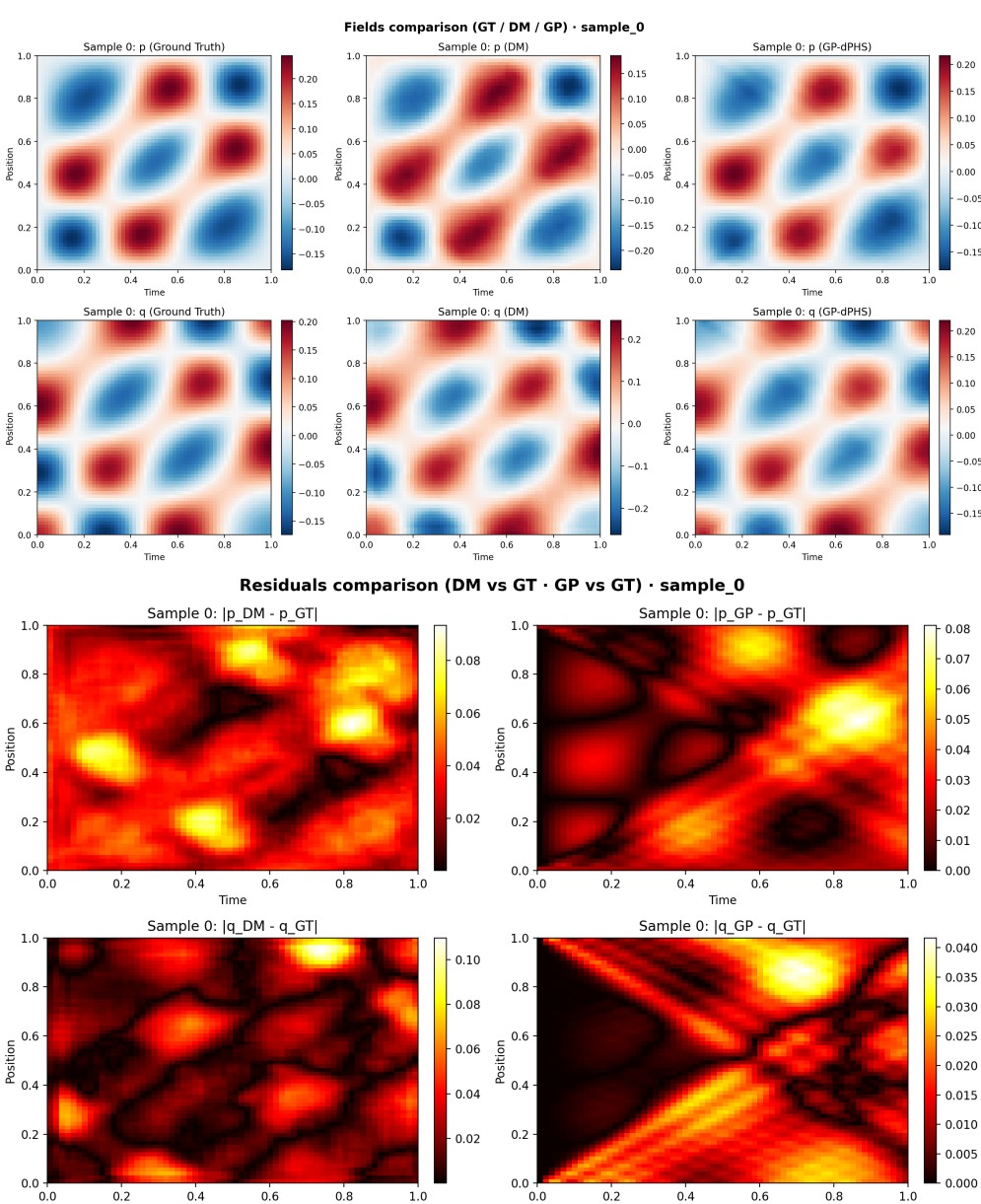

Figure 8: Validation of GP-dPHS performance and PHDME performance. The left column is the ground-truth dynamics generated by A.1, the middle column is the forecast made by the proposed method, and the last column is the GP-dPHS prediction based on the initial conditions. Key takeaway: Both GP-dPHS and PHDME have learned the correct dynamic patterns, but not 100% perfect. The red residual comparsion figures show the differences, notice that the magnitude of the residual is very low.

2. **Reconstruction of the displacement using the generated states.** Train the diffusion model in using the state and state derivatives of the system, which is the key to getting

rid of the exact function of movements. We want to validate that the proposed method can reconstruct the displacement over time by using the predicted state.

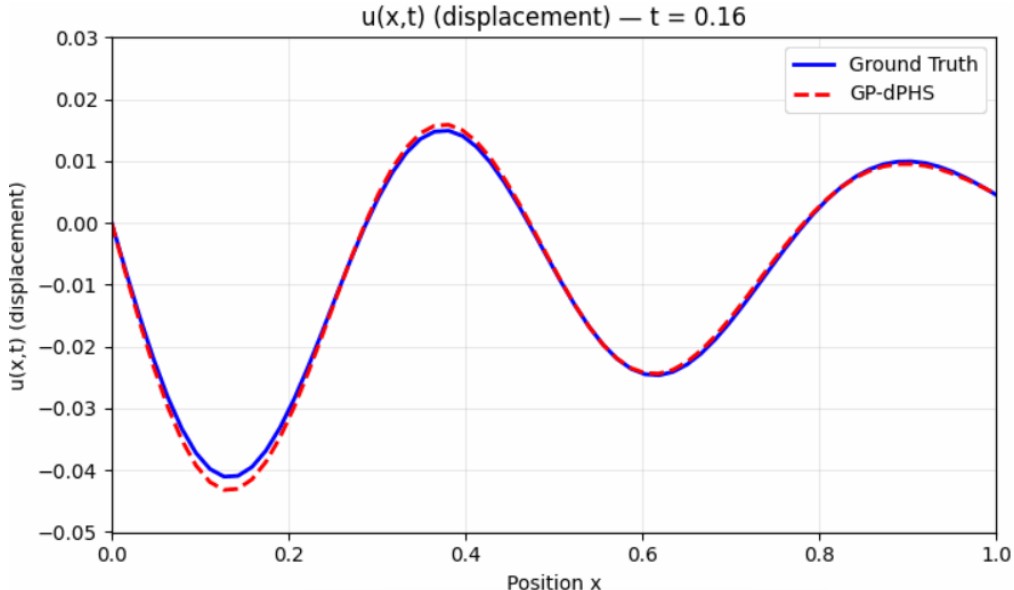

Figure 9: This is the reconstruction based on the derivative field, the blue line is the movement (displacement) of the soft string system using a faithful physics simulator. The red dot line is the one reconstructed based on the derivative field using the rollout that has been mentioned above. Key takeaway: The state and state derivative method is applicable to the physics-informed machine learning.

**Notes for exact reproducibility.** Use the saved checkpoints `model_dp_trained.pth` and `model_dq_trained.pth`; read `metadata.json` for $\Delta t$, grid size, and normalization; ensure the same boundary operator as in training; and keep the discretization identical to the equations above so that the reconstruction and training distributions match. You can run the `train_gp_phs_v35.py` file to get the GIF of the reconstruction over time.

A.7  REPRESENTATION LEARNING OF PHDME

**Instantiation of the physics-informed loss and uncertainty weighting.** Equation (4) in the main text uses an abstract physics regularizer $\widetilde{\mathcal{R}}_{\text{phys}}$ with a generic normalization factor $|\Omega|$. In our implementation, this abstract notation is instantiated as a variance-weighted residual over the discrete spatiotemporal grid and the training mini-batch.

For each denoised prediction $\widehat{\boldsymbol{A}}^{(0)} = f_\theta([\boldsymbol{A}^{(m)}, \mathbf{c}_{\text{init}}], m)$ at diffusion step $m$, the model outputs two fields on a regular one-dimensional grid,

$$p_\theta(t_i, x_j), \quad q_\theta(t_i, x_j) \in \mathbb{R}, \qquad i = 0, \ldots, N_t - 1, \ j = 0, \ldots, N_x - 1,$$

where $p_\theta$ and $q_\theta$ are the predicted momentum and strain for the string, and $N_t, N_x$ are the numbers of temporal and spatial grid points. During training we process a mini-batch of size $B$, so that the discrete domain actually used inside the code is

$$\Omega_{\text{st}} = \{1, \ldots, B\} \times \{0, \ldots, N_t - 1\} \times \{0, \ldots, N_x - 1\}, \qquad |\Omega_{\text{st}}| = BN_tN_x,$$

and the abstract factor $1/|\Omega|$ in equation (4) is implemented as an average over all elements of $\Omega_{\text{st}}$.

The GP-dPHS module exposes a Hamiltonian-based representation of the state through the learned energy gradients

$$\delta_{\boldsymbol{x}}\widehat{H}(p, q) = \begin{bmatrix} \mu_p(p, q) \\ \mu_q(p, q) \end{bmatrix},$$

where $\mu_p$ and $\mu_q$ denote the GP posterior means for $\partial E/\partial p$ and $\partial E/\partial q$, respectively. Specializing the distributed port-Hamiltonian structure to the one-dimensional string yields the continuous dynamics

$$\partial_t p(x,t) = \partial_x \mu_q\big(p(x,t), q(x,t)\big), \qquad \partial_t q(x,t) = \partial_x \mu_p\big(p(x,t), q(x,t)\big).$$

On the discrete grid, we approximate derivatives using second-order finite differences. Let $\Delta t$ and $\Delta x$ denote the time and space steps. For each batch index $b$, time index $i$, and spatial index $j$, the discrete time derivatives of the predicted fields are

$$\Delta_t p_\theta^b(i,j) \approx \frac{p_\theta^b(t_{i+1}, x_j) - p_\theta^b(t_{i-1}, x_j)}{2\,\Delta t}, \qquad \Delta_t q_\theta^b(i,j) \approx \frac{q_\theta^b(t_{i+1}, x_j) - q_\theta^b(t_{i-1}, x_j)}{2\,\Delta t},$$

with forward and backward stencils used for $i = 0$ and $i = N_t - 1$. Spatial derivatives of the GP energy gradients are defined analogously,

$$\Delta_x \mu_q^b(i,j) \approx \frac{\mu_q^b(t_i, x_{j+1}) - \mu_q^b(t_i, x_{j-1})}{2\,\Delta x}, \qquad \Delta_x \mu_p^b(i,j) \approx \frac{\mu_p^b(t_i, x_{j+1}) - \mu_p^b(t_i, x_{j-1})}{2\,\Delta x},$$

again with one-sided stencils at the spatial boundaries $j = 0$ and $j = N_x - 1$. Given these discrete operators, the local port-Hamiltonian residuals at $(b, i, j) \in \Omega_{\mathrm{st}}$ are

$$r_p^b(i,j) = \Delta_t p_\theta^b(i,j) - \Delta_x \mu_q^b(i,j), \qquad r_q^b(i,j) = \Delta_t q_\theta^b(i,j) - \Delta_x \mu_p^b(i,j).$$

Because the GP-dPHS is Bayesian, it also provides predictive variances for the energy gradients. At each point $(b, i, j)$ we obtain

$$\sigma_q^2(b,i,j) = \mathrm{Var}\big[\partial E/\partial q \mid p_\theta^b(t_i, x_j), q_\theta^b(t_i, x_j)\big], \quad \sigma_p^2(b,i,j) = \mathrm{Var}\big[\partial E/\partial p \mid p_\theta^b(t_i, x_j), q_\theta^b(t_i, x_j)\big],$$

from the GP posterior. To obtain an uncertainty measure for the spatial derivatives $\Delta_x \mu_q$ and $\Delta_x \mu_p$, the implementation propagates these variances through the finite-difference stencil. For example, the variance of the central-difference approximation to $\partial_x(\partial E/\partial q)$ at an interior spatial index $j$ is approximated as

$$\sigma_{q,x}^2(b,i,j) \approx \frac{\sigma_q^2(b,i,j{+}1) + \sigma_q^2(b,i,j{-}1)}{4\,\Delta x^2},$$

with analogous expressions for $\sigma_{q,x}^2$ and $\sigma_{p,x}^2$ at the boundaries and for the $\partial E/\partial p$ channel. These quantities are computed in the code as var_dEdq_dx and var_dEdp_dx.

The uncertainty-aware physics loss used in all string experiments is then

$$\widetilde{\mathcal{R}}_{\mathrm{phys}}\big(\widehat{\boldsymbol{A}}^{(0)}; \mathbf{c}_{\mathrm{init}}\big) = \frac{1}{|\Omega_{\mathrm{st}}|} \sum_{(b,i,j) \in \Omega_{\mathrm{st}}} \Big( w_q(b,i,j)\,\big|r_p^b(i,j)\big|^2 + w_p(b,i,j)\,\big|r_q^b(i,j)\big|^2 \Big) + \lambda_{\mathrm{bc}}\,B\big(p_\theta, q_\theta; \mathbf{c}_{\mathrm{init}}\big),$$

where $B(\cdot; \mathbf{c}_{\mathrm{init}})$ is the boundary and conditioning penalty described in Section A.5, and the weights

$$w_q(b,i,j) = \frac{1}{\sigma_{q,x}^2(b,i,j) + \varepsilon}, \qquad w_p(b,i,j) = \frac{1}{\sigma_{p,x}^2(b,i,j) + \varepsilon},$$

are inverse variances with a small numerical stabilizer $\varepsilon > 0$. In the actual implementation this expression is computed as the mean over $\Omega_{\mathrm{st}}$, that is, the factor $1/|\Omega|$ in equation (4) is concretely

$$\frac{1}{|\Omega_{\mathrm{st}}|} = \frac{1}{B N_t N_x},$$

and the per-point weights $w_p, w_q$ are derived from the GP posterior variance at each grid location.

From a representation-learning perspective, the GP-dPHS defines a Hamiltonian energy representation $\delta_{\boldsymbol{x}} \widehat{H}$ on the space of string states. The uncertainty-weighted residual above encourages the denoiser's predi

A central component of PHDME is the use of Gaussian Processes to learn the energy representation of the distributed port-Hamiltonian string system from limited data. Unlike purely data-driven models that fit trajectories directly, our GP-dPHS surrogates approximate the underlying gradients of the Hamiltonian, $dE/dp$ and $dE/dq$, providing a structured representation aligned with physical laws.

**Learning energy gradients.** The training data consist of spatiotemporal fields of momentum $p$ and strain $q$ generated from the wave system. From these, we compute integrated derivatives that serve as training targets for the GP models. Two Gaussian Processes are trained jointly: one learns the mapping $(p, q) \mapsto dE/dp$ and the other $(p, q) \mapsto dE/dq$, thereby embedding the system into an implicit energy functional. This construction encodes the Hamiltonian structure directly into the representation space.

**Visualization of learned surfaces.** Figure 10 show the learned GP surfaces for $dE/dp$ and $dE/dq$, respectively, overlaid with the training data. Even with only $1640$ training data points drawn from a single Hamiltonian-consistent trajectory, the GP recovers smooth and coherent energy gradients across the $(p, q)$ domain. This confirms that the representation is not tied to specific trajectories, but generalizes across state space.

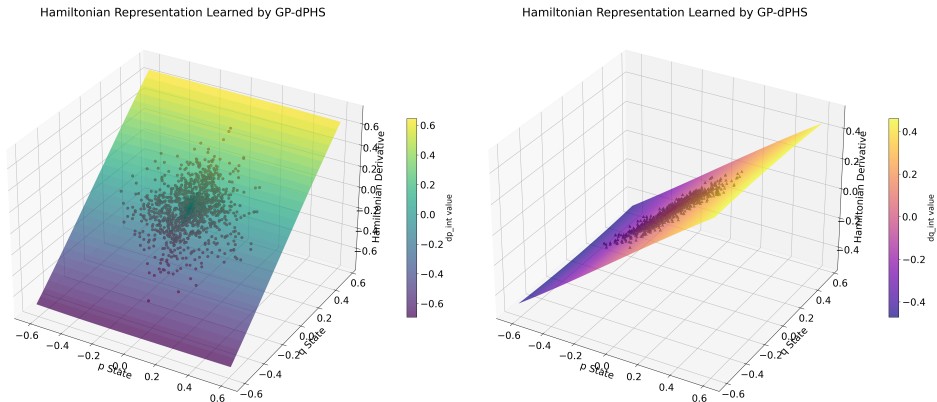

Figure 10: GP-learned representation of $dE/dp$ and $dE/dq$(partial derivative of energy) with training data points. The limited observations lie on the surface of the GP plane, indicating the correct and smooth energy representation of the system.

**Correctness of the PHDME prediction.** To verify that the proposed PHDME respects the learned Hamiltonian structure, we compute the Hamiltonian energy of both the ground–truth solution and the PHDME prediction for each generated trajectory using $H(p, q) = \int (p^2 + q^2)/2 \, dx$. Figure 11 shows a representative test sample: the energy curve of the PHDME prediction (red, dotted) almost perfectly overlaps with the ground–truth ODE solver (black, solid), stays strictly positive, and does not exhibit any artificial growth over time. Since the underlying string simulator does not include an explicit damping term, the theoretically correct behavior is energy conservation; numerically, this manifests as an energy profile that is effectively constant and at most weakly non–increasing due to discretization error. The close match between the two curves indicates that PHDME does not

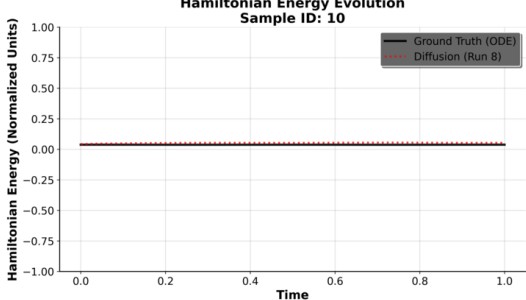

Figure 11: PHDME follows the same non-increasing Hamiltonian profile as the simulator, demonstrating adherence to the underlying physics.

inject spurious energy and its latent representation faithfully follows the same Hamiltonian law as the governing dPHS, rather than merely fitting snapshots in a purely data–driven manner.

**Role in PHDME.** These GP-learned energy gradients form the backbone of the physics-informed diffusion model. Instead of constraining the generative model with explicit PDE coefficients, PHDME leverages the GP posterior as a flexible representation of admissible energy functionals. During diffusion training, the GP structure enters the physics loss to guide denoising steps toward physically consistent dynamics. This tight coupling ensures that the learned latent dynamics reflect both data evidence and energy-based physics, enabling sharper generalization to unseen conditions.

### A.8 CONFORMAL PREDICTION, EXCHANGEABILITY, AND EMPIRICAL DIAGNOSTICS.

There are two stages of uncertainty quantification setting in proposed PHDME pipeline, one is the deep prior uncertainty based on GP-dPHS to inform the training process of the data uncertainties, the other is calibrated conformal prediction. We equip PHDME with split conformal prediction on the scalar trajectory error in order to obtain distribution–free uncertainty sets for the learned spatio–temporal representation.

**Conformal prediction based on exchangeability.** Conformal calibration is performed on a held out subset of the synthetic PDE dataset that is not used for training the diffusion model. The calibration and test subsets are constructed by random splitting of the same simulator generated corpus and are then processed by the same evaluation pipeline, so that the underlying pairs $(\mathbf{c}_{\text{init}}, \boldsymbol{A}^\star)$ are i.i.d. and hence exchangeable across both splits. In `evaluate_conformal_prediction_fast.py`, each batch provides tensors of shape $(B, 4, X, T)$; for a given initial condition the conditioning channels $[\boldsymbol{A}^{(0)}, \mathbf{c}_{\text{init}}]$ are repeated $M$ times along the batch dimension, and the optimized sampler `DenoisingDiffusionLite.p_sample_loop` is run once with i.i.d. Gaussian noise initialization of shape $(BM, 2, X, T)$, producing $M$ stochastic samples that are conditionally independent and identically distributed given the initial condition. The resulting mean squared error scores are computed over space and time for each draw, stored as a flat array of length $N \times M$. We treats these scores as an exchangeable sequence when computing overall coverage and per trajectory coverage statistics. The conformal boundary $\tau_{1-\alpha}$ is obtained beforehand by running a separate calibration script that sets $\tau_{1-\alpha}$ to the empirical $(1 - \alpha)$ quantile of the calibration scores. See 12 for the main experiment details we choose $\alpha = 0.1$ and $M = 100$, which yields $10\,000$ draws on both calibration

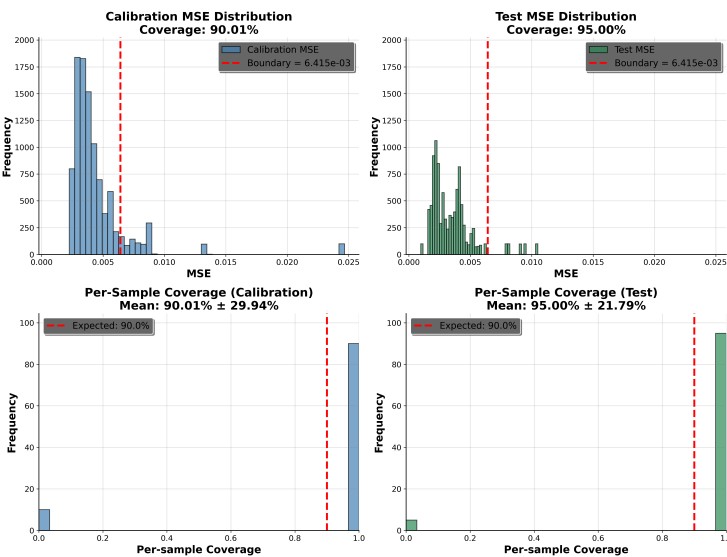

Figure 12: Calibration and test MSE distributions with the fixed conformal boundary $\tau_{1-\alpha}$. Key takeaway: The calibrated score strictly holds for an on-the-fly unseen test set, where the test set data is never seen in the calibration set.

and test sets, and the summary file reports an overall coverage of $90.01\%$ on calibration and $95.00\%$ on the held out test trajectories, indicating a slightly conservative predictor on unseen data.

**Conformal prediction coverage analysis.** Figure 13 visualizes how well our conformal prediction bands are calibrated across a range of target coverages. The horizontal axis shows the nominal coverage level $1 - \alpha$ used when constructing the bands, and the vertical axis reports the empirical coverage, that is, the fraction of trajectories whose ground-truth paths fall inside the predicted bands. The red dashed diagonal corresponds to perfect calibration, where empirical and nominal coverages coincide. Blue circles denote results on the calibration set used to fit the conformal threshold and lie almost exactly on this diagonal, confirming that the procedure is implemented correctly. Green squares show performance on a disjoint test set: the curve remains close to the diagonal and is consistently above it, indicating that our intervals are slightly conservative but never under-cover. In particular, at the target level $1 - \alpha = 0.9$ the empirical test coverage is around $0.95$, demonstrating that the conformal layer generalizes to unseen trajectories and provides reliable uncertainty quantification for PHDME forecasts.

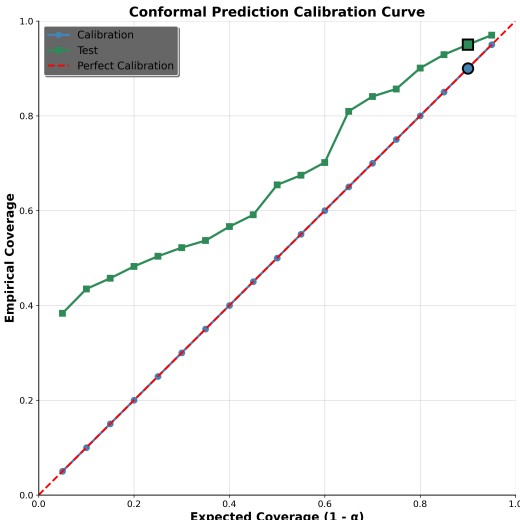

Figure 13: Take-away: The conformal prediction bands are well calibrated and slightly conservative, reliably achieving at least the desired coverage on unseen test trajectories.

## A.9 NEURALODE BASELINE ANALYSIS

In our experiments, NeuralODE (Chen et al., 2018) serves as a purely data-driven baseline that has no access to the underlying Hamiltonian or PDE structure. The model parameterizes a latent vector field and is trained only to minimize prediction error on observed trajectories, without any physics-informed regularization or constraints. As a consequence, NeuralODE can only leverage the limited set of initial conditions and time horizons present in the training split; it cannot exploit knowledge of conserved quantities or boundary conditions to interpolate or extrapolate beyond this regime. We therefore evaluate it on a more challenging setting where the test trajectories, including the real-world spring dataset, exhibit different initial displacements and modal compositions from those seen during training.

**Visualization of the trained NeuralODE.** To verify that the baseline is properly optimized, Figure 14 visualizes NeuralODE rollouts on a representative trajectory drawn from the training distribution. Each panel shows a space–time heat map of the momentum field $p$ and configuration field $q$, comparing ground truth (left) with NeuralODE predictions (right). Along both spatial and temporal axes, the predicted wave fronts, phases, and amplitudes closely match the reference solution, indicating that the latent ODE has learned a good representation of the dynamics for the specific initial conditions it was trained on. Quantitatively, this corresponds to low reconstruction error and

qualitatively smooth, coherent patterns, confirming that the failure modes discussed below are not due to underfitting.

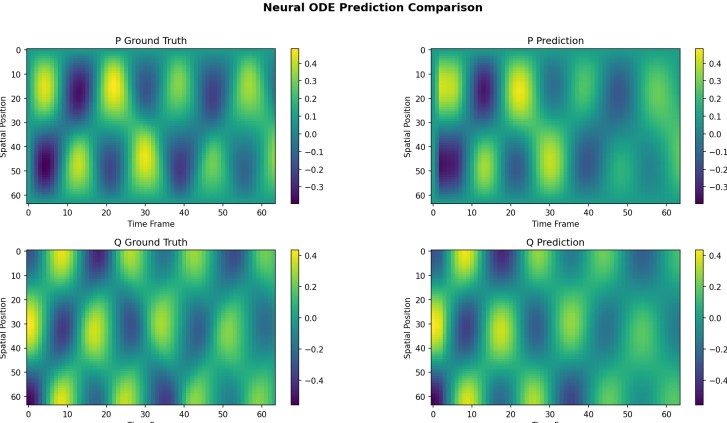

Figure 14: NeuralODE accurately reconstructs the training-distribution trajectory, with predicted p and q fields closely matching the ground-truth patterns for seen initial conditions.

**Comparison between NeuralODE and PHDME**  Figure 15 reports the same visualization for an unseen test trajectory with a different initial condition and energy level. In this regime, NeuralODE collapses: the predicted $p$ and $q$ fields quickly saturate to nearly constant values, lose the oscillatory structure present in the ground truth, and fail to capture the spatial propagation of the wave. The learned latent dynamics clearly do not generalize across initial conditions, despite performing well on the training distribution. In contrast, PHDME, shown in Figure 16, produces a rollout for the same unseen initial condition whose space–time pattern closely aligns with the ground truth in both phase and amplitude. This suggests that the Hamiltonian-informed latent representation learned by PHDME captures invariants that transfer across initial conditions, whereas the purely data-driven NeuralODE representation overfits to the finite set of observed trajectories and lacks the inductive bias needed for robust out-of-distribution generalization.

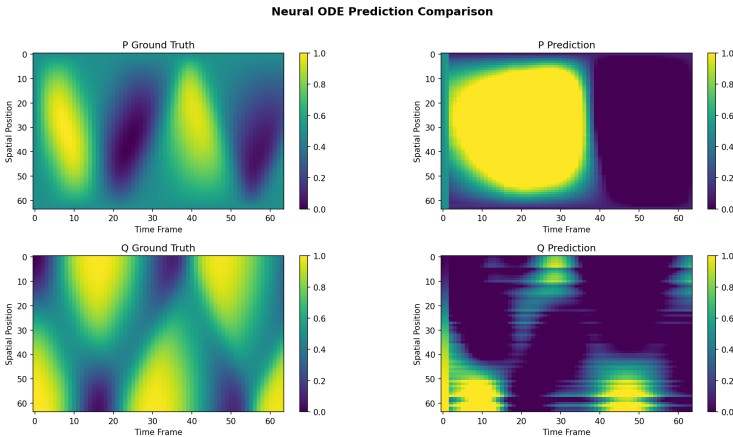

Figure 15: NeuralODE fails on the unseen initial conditions during test time. The same prediction of PHDME is on the next page.

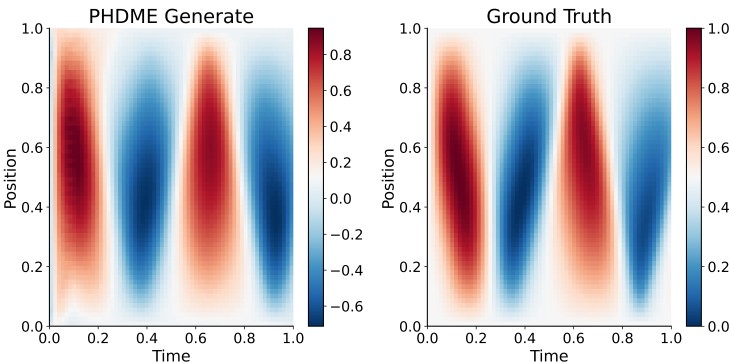

Figure 16: Credit to the physics-informed structure, the proposed PHDME makes relatively close predictions on the unseen initial condition.

### A.10 FUTHER DISCUSSION AND LIMITATIONS

**Limitations under extreme scales.** While the GP-based representation is robust to moderate data scarcity, it exhibits limitations when the dynamics evolve near extremely small state magnitudes. In these regimes, the training data provide only sparse coverage of the $(p, q)$ space, and the GP posterior surfaces tend to flatten, resulting in poor approximation of the true energy gradients. Consequently, when the diffusion model is conditioned on such representations, generated samples may fail to capture fine-scale oscillatory behavior. This effect is visible in the tails of the learned surfaces, where variance grows and predictions become less structured.

**Relative performance.** Despite these limitations, PHDME consistently outperforms non-physics baselines and ablated variants. Even when extreme scales introduce local inaccuracies, the GP-informed energy representation provides global structural regularization, preventing the generative process from drifting into unphysical states. As a result, the model produces sharper and more reliable forecasts on average, while the baselines either overfit to data trajectories or violate physical constraints. Thus, although failure cases exist at vanishingly small state magnitudes, the method achieves overall superior representation quality and downstream predictive performance.

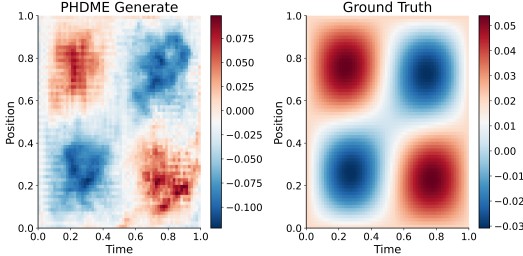

Figure 17: Under extremely small scale, the performance of the method may be compromised.

| Method | 100 samples | 1,000 samples | 10,000 samples | Avg. speed (s/sample) |
|--------|-------------|---------------|----------------|----------------------|
| GP-dPHS | 4:40 | 1:06:40 | 11:23:20 | 4.1 |
| PHDME | 20s | 3:21 | 33:23 | 0.2 |

Table 2: Generation speed comparison between GP-dPHS and PHDME. Reported time to generate different numbers of samples and the corresponding average. The PHDME is measured on a standard grid with 50 diffusion steps. while GP-dPHS is evaluated at a 640 square grid to give good derivative output. Otherwise, the long-horizon rollout of GP-dPHS would compromise the accuracy.

**Ablation: Selecting GP-dPHS as the Deep Prior.** To isolate the benefit of using a GP-dPHS as the guiding prior for the diffusion model, we conduct a controlled ablation in which we replace the GP-dPHS energy-gradient models with an oracle quadratic Hamiltonian estimator. Using the *same* training data, we perform linear regression to obtain $\partial H/\partial p$ and $\partial H/\partial q$, which corresponds exactly to fitting a global quadratic Hamiltonian. Even under this favorable assumption for the baseline, the GP-dPHS prior achieves a markedly lower MSE (0.1818 compared to 0.2967), indicating that nonparametric learning of variational derivatives provides a substantially stronger inductive bias than enforcing a fixed quadratic form. This observation aligns with the broader motivation of our method: in realistic settings, the Hamiltonian is unknown and seldom quadratic, so prescribing a closed-form energy is both restrictive and brittle. GP-dPHS instead learns a flexible representation of the underlying energy landscape, offering a more informative and generalizable deep prior for physics-informed diffusion models.

