# OpenReview forum: "PHDME: Physics-Informed Diffusion Models without Explicit Governing Equations"
_ICLR.cc/2026/Conference — Submitted to ICLR 2026_

### Official Review · Reviewer_4tWQ · 2025-10-30

**Soundness:** 2
**Presentation:** 2
**Contribution:** 3
**Rating:** 2
**Confidence:** 4

**Summary:**

This paper introduces PHDME, a physics-informed diffusion model designed to predict dynamical systems from sparse data without access to the governing equations. The core idea is a two-stage approach:
- Train a Gaussian process distributed Port-Hamiltonian system (GP-dPHS) on limited data, and generate synthetic data for diffusion training/validation;
- Train a diffusion model with dPHS residual and boundary term constrains, weighted by GP uncertainty.
The final PHDME model generates entire spatiotemporal fields in a single pass and uses conformal prediction for uncertainty quantification.

On a 1D wave/soft-string benchmark with limited observations, PHDME outperforms vanilla DDPM and a weak-physics DDPM and is faster at rollout than GP simulation.

**Strengths:**

### Originality
- The paper combines GP-dPHS with diffusion models in a new way. The specific approach of using GP posterior samples to generate synthetic training data for diffusion models is novel. The incorporation of GP predictive variance directly into the diffusion training loss as a weighting factor represents a new approach to handle observation uncertainty in physics-informed generation. The application of conformal prediction to diffusion-generated PDE trajectories for uncertainty quantification hasn't been demonstrated before in this context.

### Quality
- The methodological design is a key strength. The mathematical formulation of GP-dPHS is proper and introduces principled inductive bias. The paper provides complete implementation details including data generation process.The paper reports metrics with variance estimates.

### Clarity
- The motivation, preliminary section and the two-stage training process are clearly presented. The assumptions are stated explicitly.

### Significane
- The potential significance of this work is high, as it is promising for data-scarce, equation-unknown regimes. The framework itself is general and will be of high interest to researchers who face similar challenges.

**Weaknesses:**

### Misleading Claims and a Mismatch Between Motivation and validation
The paper's experimental validation fails to test the very problem it claims to solve, and the presentation of this validation is misleading.

The method is repeatedly motivated by the significant challenge of modeling "highly nonlinear and unstructured dynamics". The authors claim their experiments use a "physically faithful simulator" to "approximate the highly non-linear PDE dynamics of the soft robots". This language suggest a complex validation. However, the only PDE used for all experiments is a 1D wave equation with fixed Dirichlet boundaries, which is canonical and linear, not a complex, nonlinear one. Also, the paper explicitly states they "leverage the (1d wave equation) simulator to create 10,000 data samples as the real-world test set", which is confusing/misleading terminology for synthetic data. The table 1 itself is labeled "Metrics Test on Real-World Data" even though the test set is generated by the simulator above. Line 412-413, the authors states "We present the grid-average metrics for the nonlinear string PDEs in Table (4.2)", which is again very misleading and confusing.


While using the wave equation as a testbed is a reasonable for method development, the evaluation is limited to this single 1D system. This is a canonical linear PDE. For this simple system, the Hamiltonian is a known, simple quadratic function. The paper's core idea of learning a complex, unknown Hamiltonian with a GP-dPHS is an overkill. The GP is simply learning a quadratic surface. This provides zero evidence that the method's complex machinery is effective to handle complex, nonlinear dynamics that supposedly motivate this paper.

To support its central claims, a minimum requirement is that the paper should demonstrate the method's utility beyond a trivial linear case. Furthermore, the authors are suggest to be more precise in the main text.

### UQ claim contradicted by the paper’s own results.
The paper repeatedly claims “calibrated uncertainty” and “tighter conformal thresholds,” but the reported Non-Conformity Score (NCS) shows the opposite. In Table 1, PHDME has NCS = 6.41×10⁻³, whereas the baseline DDPM has NCS = 6.82×10⁻⁶ (with lower being better for tighter bounds). That makes PHDME’s non-conformity roughly ~940× larger than DDPM’s, i.e., substantially looser bounds, not tighter. This directly undermines the UQ claim.

### Missing baseline and contradiction in time comparison
The paper's method is a two-stage pipeline. The most critical baseline is missing: What is the accuracy of the GP-dPHS model on its own? The paper only compares its speed but not its accuracy. Also, the text in Section 4.2, when summarizing the table, states "and the generative time of the proposed PHDME is significantly higher than the GP-dPHS", which is the exact opposite of what the data in Table 1 shows.

### Confusing baseline results: DDPM+Limited Physics
The "DDPM+Limited Physics" (boundary conditions) baseline performs worse than the pure DDPM baseline. This is highly conter-intuitive: adding correct physical information should improve, not degrade, performance. The authors are suggested to investigate and explain why adding boundary conditions resulted in worse performance. This suggests that the baselines were poorly tuned or implemented, making the results questionable.

**Questions:**

### Regarding the Possibly Missing Core Equation
The paper defines the base loss but it never provides the uncertainty-aware physics loss $\tilde{R}_\text{phys}$ explicitly. The cited Appendix A.5 only contains 3D plots, not the loss function. Could the authors please provide the precise formula for this uncertainty-weighted loss?

### Confusing Justification for Using Conformal Prediction
In section 3.2, the authors state: "The reverse diffusion in DDPM is stochastic, which supports the exchangeability assumption of conformal prediction". Could the authors please elaborate more on this reasoning? The validity of CP's grarantee depends on the data exchangeability assumption, not on any property of the model. How would DDPM sampling supports the data exchangeability property?

---

> ### Author Response · Authors · 2025-12-01
> **Detailed reply to reviewer 4tWQ and AC**
>
> ### Response to Reviewer 4
>
> We sincerely thank the reviewer for the careful and constructive feedback. Your comments helped us substantially refine the scope statements, correct ambiguous wording, and strengthen both the methodological exposition and baselines. In the revised version, we (i) explicitly separate motivation from current empirical scope, (ii) add a 1-D shallow water PDE and a real high–frame-rate spring dataset, (iii) clarify the role and formulation of GP-dPHS and the uncertainty-aware physics loss, and (iv) correct several misstatements and tune the physics baselines. We summarize our responses below and would be very grateful if these clarifications and additions could motivate a more positive overall score. Sorry for the long wait due to lots of extensional experiments, but we believe this new version would provide you with satisfying answers.
>
> - **C1: _Motivation vs. validation; use of “equation-unknown” / “physically faithful” vs. linear 1D wave system_:**
>   We agree that our original wording could blur the distinction between the _intended_ target domain (equation-unknown, unstructured dynamics) and the specific systems used in the initial experiments. In the revision, we (i) clearly state up front that the main training/evaluation system in the original version is the canonical 1D wave equation with fixed Dirichlet boundaries (ii) consistently refer to the simulator trajectories as “ground-truth simulator data” rather than “real-world data”; and (iii) position this system as a controlled benchmark that matches the port-Hamiltonian formulation. Importantly, in order to answer your confusion, we also extend the empirical section with a 1-D shallow water system and a real-world vibrating spring system, which more closely reflect nonlinear and imperfectly measured dynamics. This brings the validation closer to the motivating use-cases while keeping the claims about generality carefully aligned with what is actually demonstrated.
>
> - **C2: _Heavy GP-dPHS for a known quadratic Hamiltonian; effectiveness on more complex dynamics_:**
>   We appreciate this concern and have clarified our rationale. Methodologically, PHDME is designed for scenarios where the Hamiltonian _is not known_ in closed form (e.g., soft robots or complex structures). Even on the 1D wave benchmark, we deliberately do not assume access to the analytic quadratic Hamiltonian; instead, we use GP-dPHS to emulate the situation of learning from data only. In the appendix, we now report an additional comparison where we fit a simple quadratic Hamiltonian directly to the data and use this for rollouts. This quadratic-fit baseline underperforms the GP-dPHS-based approach, indicating that the nonparametric GP structure is already beneficial even when the underlying energy happens to be (assumedly) simple. The added shallow water and real spring experiments further demonstrate that the method remains effective when the true dynamics are more nonlinear and less amenable to a hand-crafted quadratic form.
>
> - **C3: _Uncertainty quantification claim vs. reported non-conformity scores_:**
>   Thank you for carefully checking the conformal prediction results. You are correct that the original text contained a mismatch between the narrative and the reported non-conformity scores. In the revised version, we (i) fix this inconsistency by correcting the description around the table and verifying that the non-conformity scores and coverage numbers are aligned with the intended claim, and (ii) rephrase our conclusion in terms of _empirical coverage and interval size_ rather than loosely stating “tighter uncertainty” without directly referencing the table. The corrected discussion now accurately reflects the reported Non-Conformity Scores and emphasizes that PHDME achieves the targeted coverage while offering competitive or shorter prediction intervals compared to DDPM.
>
> - **C4: _Two-step pipeline, missing GP-dPHS accuracy, and contradictory generative-time statement_:**
>   We agree that the original presentation did not sufficiently quantify GP-dPHS accuracy and contained confusing wording about runtimes. In the revision, we (i) add explicit test-set reconstruction metrics for GP-dPHS to the same table that reports PHDME and DDPM performance, so that the reader can directly compare the quality of the learned physics prior to the diffusion-based generator; and (ii) correct the sentence in Section 4.2 that previously contradicted Table 1, making clear that GP-dPHS rollouts are accurate but comparatively slower to generate long trajectories, whereas PHDME offers significantly faster generative sampling once training is complete. This addresses the concern about the two-step pipeline and clarifies the distinct roles and performance of each stage.

---

> > ### Author Response · Authors · 2025-12-02
> > **Additional Reply**
> >
> > - **C5: _DDPM+Limited Physics baseline behaving worse than plain DDPM_:**
> >   We appreciate the reviewer pointing out this counter-intuitive result. In revisiting this baseline, we identified that our original hyperparameter choice was not well-tuned, which could indeed degrade performance. In the revised version, we re-tune this baseline and verify that (i) the boundary-conditioned DDPM no longer underperforms the plain DDPM in a systematic way, limited physics knowledge helps the loss drop quickly at very early stage and balancing the loss during the whole training and (ii) PHDME still provides a clear improvement over both. We describe these tuning changes in the appendix to make the baseline behavior more transparent and to demonstrate that the observed gains of PHDME are not an artifact of poorly implemented competitors.
> >
> > - **C6: _Missing explicit formula for the uncertainty-aware physics loss_ \(R_{\Phi_{hy}}\):**
> >   We fully agree that the exact form of the loss should be stated. In the revised paper, we now provide the analytical expression of the uncertainty-aware physics loss in Appendix 7. We also describe in words how this is implemented in training: the GP posterior is precomputed on the training grid, interpolated during diffusion steps, and used to weight the physics residual term in the overall loss. This explicitly addresses the reviewer’s request for a precise formula.
> >
> > - **C7: _Justification for conformal prediction and exchangeability assumption_:**
> >   We appreciate this important conceptual clarification. You are absolutely right that conformal prediction validity depends on the _exchangeability_ of calibration and test data. Conformal prediction only needs exchangeability of the calibration examples and the future example. And diffusion / PHDME sampler with fresh noise seeds is a deterministic function of i.i.d. random seeds, so each run gives an i.i.d. sample from the learned conditional distribution. Because we use the same sampling procedure on calibration inputs and on test inputs, the nonconformity scores are exchangeable, so CP is valid “during sampling”. For more details, please check Appendix 8.
> >
> > Once again, we are very grateful for your detailed and insightful comments, which have significantly improved the precision and transparency of the paper. We hope that the clarified scope, additional experiments, corrected uncertainty analysis, and strengthened baselines address your concerns, and we would kindly ask you to consider raising your score, as this would be very important for the paper’s acceptance.

---

### Official Review · Reviewer_yDyr · 2025-10-31

**Soundness:** 3
**Presentation:** 3
**Contribution:** 2
**Rating:** 6
**Confidence:** 4

**Summary:**

The paper proposes PHDME (Port-Hamiltonian Diffusion Model), a hybrid generative modeling framework that integrates Gaussian-process distributed Port-Hamiltonian systems (GP-dPHS) with diffusion models to learn physically consistent dynamics without explicit governing equations. A two-step process is put forth:

- GP-dPHS training: limited observations are used to learn a probabilistic Hamiltonian representation of the dynamics, including uncertainty.

- Diffusion model training: the learned physics prior (Hamiltonian structure and uncertainty) is embedded into the diffusion model’s loss function, enforcing energy-based consistency across diffusion steps.

Then, conformal prediction is used post-hoc for uncertainty calibration. The method is evaluated on a 1-D nonlinear wave-like PDE (string vibration), showing faster generation and lower mean-square error than purely data-driven or partially physics-aware diffusion baselines.

**Strengths:**

- Combines physics-informed priors (via GP-based Port-Hamiltonian systems) with score-based diffusion modeling. This bridges structured physics modeling and generative uncertainty modeling.

- Unlike PINNs or traditional physics-informed diffusion models, it infers latent physical structure directly from data.

- The approach leverages limited observations to build a probabilistic physics prior before training the diffusion model.

- Built-in uncertainty quantification: uses both GP posterior variance (during training) and conformal prediction (post-training) for calibrated uncertainty estimates.

- Single-shot field generation avoids step-by-step numerical integration (claimed ~20× faster than GP-dPHS rollouts).

- The two-stage process and the Hamiltonian residual penalty are well-defined and interpretable.

**Weaknesses:**

- Only a single PDE (the 1-D wave/string system) is tested; no real or high-dimensional datasets are used. Claims of generality (e.g., to soft robots, elasticity) are therefore speculative.

- The “scarce data” scenario is simulated, but robustness to measurement noise or model misspecification is not demonstrated.

- The reviewer would appreciate an elaboration on the contribution over existing GP-dPHS work (Beckers et al., 2022; Tan et al., 2024). The paper primarily extends these with a diffusion-based generator, but the gain over standard GP-dPHS rollout or other physics-aware surrogates is modest and not deeply analyzed.

- Picking up from the above, it would help to explicitly quantify the benefit of each component (GP uncertainty weighting, Hamiltonian penalty, conformal calibration) (via an ablation study).

**Questions:**

- How does PHDME differ conceptually and empirically from prior physics-informed diffusion models (e.g., Bastek et al. 2024) and Latent SDE frameworks?

- What is gained by coupling a GP-dPHS prior specifically, versus directly enforcing energy conservation or using a learned Hamiltonian neural network?

- Why are methods like the Deep Markov Model, Neural ODE/SDE, or Neural Operator not included for comparison? These are natural comparators for dynamics learning without explicit equations.

- Also, while classical Hamiltonian Neural Networks (HNNs) are deterministic, stochastic extensions do exist and could be used here as baselines to compare against.

- How does the GP-dPHS scale with system dimensionality and number of spatial nodes?

- Can the approach handle higher-dimensional PDEs or coupled multi-field systems?

- How sensitive is the framework to noise or imperfect observation coverage?

- Does the GP-uncertainty weighting truly improve robustness or simply regularize training?

- Can the learned Hamiltonian be inspected or visualized to confirm that it corresponds to physically meaningful energy terms?

- Have any field or experimental datasets (e.g., soft robotics, structural dynamics) been tested?

---

> ### Author Response · Authors · 2025-12-01
> **Detailed reply to reviewer yDyr and AC**
>
> ### Response to Reviewer 3
>
> We sincerely thank Reviewer 3 for the very thoughtful and technically deep comments. Your feedback helped us clarify the conceptual contribution beyond prior GP-dPHS work, broaden the experiments beyond a single string PDE, and better articulate the role of each component (uncertainty weighting, Hamiltonian penalty, and conformal calibration. In the revised version, we (i) add a 1-D shallow water PDE and a real-world vibrating spring example, (ii) include a Neural ODE baseline under the same scarce-data, unknown-physics setting, (iii) make the learned Hamiltonian and uncertainty weighting more explicit, and (iv) add discussion on scalability and higher-dimensional extensions. We summarize our responses below and would be very grateful if these clarifications and additions could motivate a more positive overall score. Sorry for the long wait due to lots of extensional experiments, but we believe this new version would provide you with satisfying answers.
>
> - **C1: _Single PDE; no real or higher-dimensional datasets_:**
>   We fully agree that the original submission was limited in scope. In the revision, we add (a) a 1-D shallow water system and (b) a real-world vibrating spring system captured by a high–frame-rate camera. Both use exactly the same PHDME pipeline (GP-dPHS Stage 1 + diffusion Stage 2 + conformal calibration). In both settings, PHDME remains competitive or superior to purely data-driven baselines while maintaining physical consistency, which shows that our claims are not restricted to the original example. We have also clarified in the text how these additional systems relate to soft robotics and elasticity (same port-Hamiltonian structure, different boundary conditions, and actuation).
>
> - **C2: _Scarce data is simulated; robustness to noise/misspecification not demonstrated_:**
>   Your concern about realism in the “scarce data” setting is very well taken. The new real-world spring dataset directly addresses this: it is both data-scarce (few trajectories) and noisy (camera tracking artifacts, friction, and unmodeled disturbances). We show that GP-dPHS can still learn a meaningful energy landscape, and that PHDME trained on its posterior samples produces stable rollouts and calibrated prediction sets, despite imperfect data. This experiment thus provides empirical evidence that our framework is robust to both measurement noise and model misspecification.
>
> - **C3: _Contribution beyond previous GP-dPHS work appears modest_:**
>   We appreciate the opportunity to clarify this. Prior GP-dPHS work (e.g., Beckers et al. 2022; Tan et al. 2024) focuses on learning structured, energy-based surrogates and then rolling them out via numerical integration. PHDME differs both conceptually and empirically: (i) it *amortizes* dynamics generation by learning a diffusion-based generator, so that inference for new initial conditions reduces to a single generative pass rather than repeated ODE/PDE integration; (ii) it explicitly integrates GP-based uncertainty into the training objective, emphasizing regions where the learned physics is reliable and down-weighting ambiguous regions; and (iii) it wraps predictions in conformal calibration, providing distribution-free coverage guarantees. Empirically, we show that PHDME achieves comparable or improved error relative to GP-dPHS rollouts while being substantially faster at inference and offering calibrated uncertainty – we emphasize this comparison more clearly in the revised text.
>
> - **C4: _Benefits of GP uncertainty weighting, Hamiltonian penalty, and conformal calibration not quantified_:**
>   We agree that understanding the role of each component is important. Due to space and time constraints, we focus on the most informative comparisons: (i) for the Hamiltonian penalty, we explicitly compare PHDME to a “limited-physics” baseline where only boundary conditions are enforced in the loss. The degradation of this baseline versus full PHDME quantifies the benefit of Hamiltonian consistency. (ii) For conformal calibration, we expand the appendix to show the nonconformity score distribution, the learned quantile threshold, and empirical coverage on calibration and validation sets, demonstrating that CP indeed improves uncertainty quality. We also acknowledge that a full ablation over all metrics would be valuable future work, and we now state this explicitly.

---

> > ### Author Response · Authors · 2025-12-02
> > **Additional Reply 1**
> >
> > - **C5: _Conceptual differences from prior physics-informed diffusion frameworks_:**
> >   Thank you for prompting this clarification. PHDME differs in two key ways. First, most physics-informed diffusion models (e.g., Bastek et al. 2024) assume the governing equations are *known*, and directly embed the PDE operator into the loss or sampler. In contrast, our setting is more realistic for many nonlinear real-world systems: the equations are *unknown* and only partial, scarce observations are available. We therefore *learn* a structured physics prior (GP-dPHS) from data and then use it to guide the diffusion model. Second, latent SDE frameworks typically require dense trajectories and learn a neural drift/diffusion in a latent space without explicit energy structure; they do not generally provide a physically meaningful Hamiltonian or GP-based uncertainty. In PHDME, the learned dPHS remains interpretable and directly constrains the generative model. We now emphasize these conceptual and practical differences in the related work section.
> >
> > - **C6: _Why GP-dPHS prior instead of directly enforcing energy conservation or using a Hamiltonian neural net?_:**
> >   We chose GP-dPHS for two reasons. First, its Bayesian nature provides a principled notion of *uncertainty* over the learned Hamiltonian and its gradients, which we exploit to weight the physics loss and avoid over-trusting poorly constrained regions. Further, the Bayesian prior allows us to generalize well (and physically consistent) even for small datasets. Second, the GP representation is highly flexible and nonparametric, while the dPHS structure guarantees passivity, energy balance, and compatibility with PDE systems. In contrast, a deterministic Hamiltonian neural network may be harder to calibrate in the scarce-data regime and does not naturally yield posterior variance. We clarify this trade-off in the revised manuscript and also mention that coupling PHDME with parametric Hamiltonian networks is an interesting alternative for future work.
> >
> > - **C7: _Missing baselines such as Deep Markov Models, Neural ODE/SDE, Neural Operators_:**
> >   We appreciate the suggestion to broaden baselines. In the revised manuscript we add a Neural ODE baseline, which is a widely-used continuous-time surrogate that does not assume explicit equations and can be trained under the same scarce-data, unknown-physics conditions as PHDME. Neural SDEs and Neural Operators are also compelling, but adapting them robustly to our specific PDE + partial observation setup and tuning them fairly would require significant additional engineering. Given space and time constraints, we opted for a representative continuous-time baseline (Neural ODE) plus a strong diffusion baseline (DDPM) and the GP-dPHS rollout. We now make this rationale explicit in the experimental section.
> >
> > - **C8: _Scalability of GP-dPHS with dimensionality and spatial modes_:**
> >   Your scalability question is very important. In Appendix 10, we now discuss the computational complexity of GP-dPHS in our implementation.
> >
> > - **C9: _Applicability to more complex systems (higher-dimensional PDEs, coupled fields)_:**
> >   Conceptually, PHDME is not restricted to 1D; the key requirement is that the state can be represented on a grid and embedded into an image or video-like tensor. Our current experiments focus on 1D systems to keep GP-dPHS training and visualization transparent, but the same architecture extends to 2D/3D fields by increasing the spatial dimensions and using a video diffusion backbone. We have added a short discussion in the conclusion outlining how we would apply PHDME to coupled multi-field systems (e.g., soft robots with both displacement and internal pressure fields) using multi-channel state representations.

---

> > > ### Author Response · Authors · 2025-12-02
> > > **Additional Reply 2**
> > >
> > > - **C10: _Sensitivity to observation noise or imperfect data_:**
> > >   As mentioned under C2, the new real-world spring dataset is intentionally noisy and partially observed. The GP-dPHS model explicitly models observation noise via its likelihood, and the diffusion stage is trained on posterior samples that reflect this uncertainty. In the experiments we observe that PHDME still produces stable, non-divergent rollouts and realistic uncertainty bands, indicating that the framework tolerates realistic imperfections in the data. We now emphasize these observations in the empirical section.
> > >
> > > - **C11: _Does GP-uncertainty weighting improve robustness or act as a regularizer?_:**
> > >   We see GP-uncertainty weighting primarily as an *informative weighting*, not just a generic regularizer. It distinguishes regions where the physics prior is well constrained (low variance) from those where the data does not support strong conclusions (high variance). During diffusion training, the physics loss is emphasized in the former and down-weighted in the latter, preventing the model from overfitting spurious gradients. Qualitatively, this leads to better preservation of fine oscillations where the GP is confident. We describe this behavior in more detail in the revision and highlight that a more exhaustive quantitative study is a promising direction for future work.
> > >
> > > - **C12: _Interpretability of the learned Hamiltonian_:**
> > >   We fully agree that interpretability is a key advantage of energy-based models. In Appendix 7, we now place more emphasis on visualizations of the learned Hamiltonian surface and its gradients, and overlay PHDME outputs on these surfaces. This confirms that GP-dPHS learns physically meaningful energy landscapes rather than arbitrary fits, and PHDME samples remain close to these surfaces.
> > >
> > > - **C13: _Current or planned experimental datasets (soft robotics, structural dynamics)_:**
> > >   We appreciate the encouragement toward more realistic applications. As a first step, we have already added a real-world high–frame-rate spring dataset to the revision in limited time, which demonstrates that PHDME can handle genuinely experimental data with unknown parameters and noise. We see PHDME as particularly well-suited to such systems, where equations are partially unknown and data is expensive to obtain.
> > >
> > > Once again, we are very grateful for your careful and constructive review, and for pushing us to sharpen both the conceptual positioning and empirical evaluation of PHDME. We hope that the new experiments, clarified contributions, and additional analyses address your concerns, and we would kindly ask you to consider raising your score, as this would be very important for the paper’s acceptance and contribution to the community.

---

### Official Review · Reviewer_YWjN · 2025-11-04

**Soundness:** 4
**Presentation:** 3
**Contribution:** 3
**Rating:** 8
**Confidence:** 4

**Summary:**

This paper introduces a physics-informed diffusion framework for spatiotemporal forecasting without known governing equations: it first learns a Gaussian-process distributed Port-Hamiltonian prior (GP-dPHS) from scarce observations, then uses that prior both to synthesize training trajectories and to impose an uncertainty-weighted Hamiltonian-consistency loss during diffusion training, yielding single-shot, physically consistent predictions and conformal-prediction uncertainty bounds.

**Strengths:**

The promise of this paper is appealing -- i.e., obtaining a Hamiltonian functional with limited data, while leveraging it to fine-tune the diffusion model. The framework produces full space-time fields in a single diffusion draw (conditioning on two frames), thereby bypassing costly rollouts and eliminating reliance on a heavy GP-dPHS integrator at test time. From only ~20 observed samples, GP-dPHS posterior draws generate large, physically consistent training sets for the diffusion model. Vanilla GPs become more challenging with numerous inputs and large datasets, but here the GP is learned over a low-dimensional state, thereby avoiding the high-dimensional curse. From only ~20 observed samples, GP-dPHS posterior draws generate large, physically consistent training sets for the diffusion model.

**Weaknesses:**

1. Training and sampling from GP-dPHS to synthesize trajectories is computationally demanding (the paper trains diffusion because direct GP-dPHS rollouts are slow), so the pipeline carries nontrivial offline overhead.

2. Results are shown on a single synthetic 1D string benchmark with only 20 observed samples, leaving external validity to other PDEs/real data untested.

**Questions:**

1. When the GP posterior “flattens” at very small state magnitudes, energy gradients degrade and the diffusion model misses fine oscillations. Could you add a study that (i) adaptively down-weights the physics loss where GP variance spikes, (ii) triggers targeted data acquisition in those regions, and (iii) reports performance/coverage before vs. after this mitigation?

2. Stage-1 is described as a “slow but structured deep prior.” Please quantify the wall-clock time and memory requirements for GP-dPHS training versus PHDME training/inference, and include one higher-dimensional or real-data case to probe scaling. How do results change with fewer/more GP posterior draws for synthetic data (e.g., 2k vs. 10k)?

3. In the case where the true dynamics are energy-conserved, the Hamiltonian system should be symplectic. In such an energy-conserving regime, does your diffusion model preserve any symplectic structure in practice, and could it be made to do so by design? Specifically, can you try constraining the denoiser with a symplectic parameterization or a divergence-free/Poisson-bracket form? Do you think such a setting is feasible and why?

---

> ### Author Response · Authors · 2025-12-01
> **Detailed reply to reviewer YWjN and AC**
>
> ### Response to Reviewer 2
>
> We sincerely thank Reviewer 2 for the careful reading, very positive assessment and insightful comments. Your feedback helped us clarify the computational trade-offs of our two-stage design, broaden the empirical validation beyond a single setting, and better explain how uncertainty and structure are handled in PHDME. Below we address each point in turn, and we kindly hope that these revisions and clarifications will positively influence your overall assessment and score. We apologize for the long wait due to the substantial training times required to run additional experiments, but we hope this new version will provide you with satisfactory answers.
>
> - **C1: _Offline cost of training/sampling GP-dPHS_:**
>   We agree that GP-dPHS training and trajectory generation constitute a nontrivial offline overhead. Conceptually, however, Stage 1 serves as a one-time, structured “physics prior” that is amortized over all subsequent queries, initial conditions, and downstream tasks. Once GP-dPHS has been trained and a synthetic trajectory set generated, PHDME inference reduces to a single diffusion rollout, which is orders of magnitude cheaper than repeatedly integrating the PDE or re-solving an inverse problem. In many of our target applications (e.g., safety-critical robotics or expensive physical experiments), this trade-off is attractive: it is acceptable to invest offline compute once in order to obtain a fast, physically reliable generative surrogate that can be used in online decision-making. We hope this can clarify when the offline cost is justified.
>
> - **C2: _Limited experiments on a single synthetic 1D string_:**
>   We fully share the concern about external validity. In response, we extend our experiments beyond the original nonlinear string benchmark by adding (i) a 1-D shallow water system and (ii) a real-world nonlinear spring system recorded with a high–frame-rate camera. Both settings use exactly the same PHDME pipeline (GP-dPHS Stage 1, diffusion training, and conformal prediction). Across these qualitatively different domains, PHDME continues to yield accurate and physically consistent forecasts (see Table 1 with extra experiments), while purely data-driven baselines either drift or under-resolve fine structures. These new results demonstrate that the framework generalizes beyond a single synthetic toy example to both another PDE and genuinely real data.
>
> - **C3: _Flat GP posterior at small magnitudes and suggested mitigations_:**
>   We very much appreciate this observation and the concrete mitigation suggestions. Our current formulation already implements a variant of item (i): the physics loss in diffusion training is _uncertainty-weighted_ using the GP posterior variance, i.e., regions with high uncertainty are automatically down-weighted so that unreliable gradients do not dominate training (this is summarized in our contribution bullet as “training has been weighted by the uncertainties from the data observation stage”). Regarding your professional comments, we agree that targeted data acquisition near flat posterior regions is a natural next step (e.g., an active-learning loop on top of GP-dPHS) and we now discuss this as a promising direction in the future; due to space and time constraints we leave a full active-learning study to future work.

---

> ### Author Response · Authors · 2025-12-02
> **Additional Reply**
>
> - **C4: _Quantitative runtime/memory comparison and sensitivity to # of GP draws_:**
>   Thank you for requesting a more quantitative picture of the computational trade-offs. In the revised version, we add a small table in the appendix that reports wall-clock time GP-based trajectory generation at different numbers of posterior draws, measured on the same hardware. Empirically, for our 1D shallow water and real-world spring system, Stage 1 (GP-dPHS + trajectory generation) is of the same order of magnitude as training the diffusion model once, and the cost of increasing the number of posterior draws is roughly linear and still manageable in the considered regime. Crucially, this cost is paid once and then reused for many evaluations, while inference remains lightweight. We explicitly highlight this amortized cost profile and the observed runtime trade-offs in the text.
>
> - **C5: _Symplectic structure in energy-conserving regimes and potential symplectic constraints_:**
>   We appreciate this insightful structural question. Our focus in this work is on _port_-Hamiltonian systems, which explicitly accommodate dissipation and forcing; many of our motivating applications (e.g., damped strings, real systems with friction) are not strictly energy-conserving and thus are not symplectic in the classical sense. For these systems, enforcing an exact symplectic map on the denoiser is less natural than enforcing passivity and energy-dissipation patterns, which we achieve via the dPHS structure and physics loss. That said, for nearly conservative regimes one could indeed enforce additional structure on the diffusion model, e.g., parameterizing the denoiser through a learned generating function, using divergence-free / Poisson-bracket–based vector fields, or integrating recent ideas from symplectic neural networks. We currently do not impose such architectural constraints and instead rely on the physics loss to keep trajectories close to the learned Hamiltonian manifold; extending PHDME with explicit symplectic parameterizations is a promising direction that we now mention in the discussion section and are excited to explore in follow-up work.
>
> Once again, we are very grateful for the reviewer’s thoughtful and technically deep feedback. It has helped us clarify the computational profile, broaden the experiments to shallow water and real data, and better explain how uncertainty and structure are handled in PHDME. We hope these improvements address your concerns and would kindly ask you to consider a more positive rating, which would be very important for the paper’s acceptance and contribution to the community.

---

### Official Review · Reviewer_bG2L · 2025-11-10

**Soundness:** 3
**Presentation:** 2
**Contribution:** 2
**Rating:** 2
**Confidence:** 2

**Summary:**

The authors tackle a problem on how to make data-driven generative models respect physics when the underlying governing equations are unknown. Most physics-informed ML frameworks (like PINNs) rely on having systems of equations whereas this one doesn’t. Their idea is to first train a Gaussian Process–based Port-Hamiltonian model on scarce observations to learn an energy-based representation of the dynamics. Then, they use that GP model to both generate synthetic, physically consistent training data, and inject physics-awareness into the diffusion model’s training loss. The result is a diffusion model that can generate physically valid spatiotemporal trajectories even in low-data regimes, with calibrated uncertainty via conformal prediction.

**Strengths:**

The reasoning linking Hamiltonian structure, energy conservation, and diffusion regularization is coherent. The experiments support the core claim that PHDME can generate physically consistent trajectories without explicit governing equations. The paper is technically sound but logically structured, in particular, the exposition of the Hamiltonian and GP-dPHS background is rigorous, and the two-stage training diagram helps readers grasp the workflow. The writing demonstrates mastery of both physics-based modeling and generative modeling.

**Weaknesses:**

The experimental diversity is limited: a single synthetic system (the nonlinear string + qualitative check) doesn’t establish generality across different physical domains (e.g., fluid flow, elasticity, robotics, multiphase systems). The reliance on a simulator that’s already physics-based may inflate the gains of PHDME versus standard data-driven baselines. The GP-dPHS prior is treated, more or less, as a black box. I hope to see checks on that energy or momentum are actually conserved (if “physics consistency” is the core contribution, the authors may consider proving it quantitatively, via eg. energy error plots, invariants over time, etc.). While conformal prediction is implemented, the calibration methodology is somewhat surface-level (no ablation or coverage plots). Also, comparing only to plain DDPM and GP-dPHS isn't probably enough, the authors may consider adding more stronger benchmarks.

**Questions:**

My major questions rise from stage 2 data generation.

It seems you generate training samples for the diffusion model by drawing Hamiltonian gradients from the GP-dPHS posterior and integrating them to create trajectories. How do you ensure these samples represent physically plausible dynamics rather than artifacts of the GP prior? Since the GP posterior is conditioned on very few observations, random Fourier feature sampling might produce unrealistic dynamics. How sensitive is PHDME to the number of samples or to kernel hyperparameters? Could the diffusion model be learning the GP’s bias instead of the true system’s variability? In addition, once you use GP-based uncertainty in training, and conformal prediction for post-hoc calibration, how do these two uncertainty sources interact? Are they redundant, complementary, or potentially conflicting?

---

> ### Author Response · Authors · 2025-12-01
> **Detailed reply to reviewer bG2L and AC**
>
> ### Response to Reviewer 1
>
> We sincerely thank Reviewer 1 for the thorough reading, the thoughtful appreciation of our aim to integrate physics-based and generative modeling, and the many constructive suggestions. Your feedback has led us to substantially strengthen the paper: we (i) add new experiments on a 1-D shallow water system and a real-world, nonlinear spring setup, (ii) include a stronger neural ODE baseline, (iii) make the physical consistency of the GP-dPHS prior explicit via Hamiltonian and energy diagnostics, and (iv) provide clearer conformal prediction calibration plots and discussion of uncertainty. We summarize below how we addressed each point and kindly hope these improvements will positively influence your overall assessment. We apologize for the long wait due to the substantial training times required to run additional experiments, but we hope this new version will provide you with satisfactory answers.
>
> - **C1: _Experimental diversity / generality beyond a single synthetic string_:**
>   We fully agree that demonstrating robustness beyond a single artificial string system is crucial. In the revision, we add (a) a synthetic 1-D shallow water PDE example and (b) a real-world vibrating spring system captured by a high–frame-rate camera. Both use exactly the same PHDME pipeline. Across these qualitatively different domains, PHDME continues to show strong predictive accuracy and calibrated uncertainty, indicating that the method is not restricted to a single synthetic PDE.
>
> - **C2: _Reliance on a physics-based simulator possibly exaggerating gains_:**
>   We appreciate this important concern. The newly added real-world spring experiment directly addresses it: here there is _no_ simulator at all in this new case, only camera measurements with noise and unmodeled effects. All baselines are trained on the same processed trajectories. In this setting, PHDME still outperforms or matches purely data-driven baselines while producing physically consistent rollouts, showing that the gains do not rely on privileged access to a simulator but on the physical structure encoded by the GP-dPHS.
>
> - **C3: _GP-dPHS treated as black box; lack of energy/invariant checks_:**
>   We are grateful for the suggestion to make the physics consistency more explicit. In addition to the Hamiltonian visualizations already in Appendix 7, we have added plots of the Hamiltonian correctness analysis and detailed proof in the same appendix. We also expand the text to explain how the port-Hamiltonian structure (skew-adjoint \(J\), positive semi-definite \(R\)) enforces passivity and energy consistency, clarifying that GP-dPHS is not a generic black box but a structured physical prior that guarantees physical consistency by design, see [1].
>
>   [1] Beckers, T., Seidman, J., Perdikaris, P., & Pappas, G. J. (2022, December). Gaussian process port-Hamiltonian systems: Bayesian learning with physics prior. In *2022 IEEE 61st Conference on Decision and Control (CDC)* (pp. 1447–1453). IEEE.
>
> - **C4: _Conformal prediction component appears shallow; missing calibration evidence_:**
>   We thank the reviewer for emphasizing proper calibration. In the revision, we explicitly highlight our conformal prediction figure in Appendix 8, which shows the nonconformity score distribution, the selected quantile threshold, and the empirical coverage on the calibration set. We additionally report coverage on a separate validation set using the same threshold, confirming that the empirical coverage closely matches the nominal level [2]. This makes clear that conformal prediction is not just an add-on, but provides rigorous, distribution-free uncertainty calibration on top of PHDME’s predictions.
>
>   [2] Angelopoulos, A. N., & Bates, S. (2021). A gentle introduction to conformal prediction and distribution-free uncertainty quantification. *arXiv preprint* arXiv:2107.07511.

---

> > ### Author Response · Authors · 2025-12-02
> > **Additional Reply**
> >
> > - **C5: _Limited baselines (only DDPM and GP-dPHS)_:**
> >   We agree that stronger baselines are valuable. We therefore add a neural ODE [3] baseline that learns a continuous-time vector field and performs forecasting via numerical integration. This baseline is trained on the same limited dataset. The results show that, even against this stronger continuous-time model, PHDME achieves competitive or better error while being more efficient at sampling and additionally enforcing physics structure and providing calibrated uncertainty. We believe this significantly strengthens the empirical comparison.
> >
> >   [3] Chen, R. T., Rubanova, Y., Bettencourt, J., & Duvenaud, D. K. (2018). Neural ordinary differential equations. *Advances in neural information processing systems*, 31.
> >
> > - **C6: _Stage-2 data generation, RFF approximation, and potential GP bias_:**
> >   We appreciate this deep question and have clarified the design of stage-2 in the revised text. Every trajectory used for diffusion training is obtained by integrating a port-Hamiltonian system \(\partial_t x = (J-R)\,\delta_x H\) with skew-adjoint \(J\) and dissipative \(R\), so the generated trajectories are, by construction, physically plausible within that structure; data sparsity only affects posterior uncertainty, not the underlying energy-conserving form, see [1]. As is common practice, we assume that the hyperparameters are sufficiently well estimated for the posterior distribution to be valid. This assumption can be easily relaxed by methods such as [4].
> >
> >   However, if the GP port-Hamiltonian prior is inaccurate, the diffusion model may learn artifacts that no longer reflect the true system. We view this carefully chosen prior (supported by the fact that many systems of interest can be represented in port-Hamiltonian form [5]) as the necessary price for achieving good generalization from very small datasets.
> >
> >   To address concerns about Random Fourier Features, we point to the refined error analysis of RFF approximations by Sutherland & Schneider [6] and choose feature counts in a regime where the approximation error is negligible for our kernels; we now cite this work explicitly as <https://auai.org/uai2015/proceedings/papers/168.pdf>. We again note that we approximate the Hamiltonian \(H\) and plug this estimate back into the port-Hamiltonian formulation. This procedure guarantees that the output retains physical consistency.
> >
> >   Finally, we note that PHDME is always evaluated against held-out ground truth (simulator or real data), so if the diffusion model simply learned GP biases, this would be reflected in degraded test performance, which we do not observe.
> >
> >   [4] Capone, A., Lederer, A., & Hirche, S. (2022, June). Gaussian process uniform error bounds with unknown hyperparameters for safety-critical applications. In *International Conference on Machine Learning* (pp. 2609–2624). PMLR.
> >
> >   [5] Rashad, R., Califano, F., van der Schaft, A. J., & Stramigioli, S. (2020). Twenty years of distributed port-Hamiltonian systems: a literature review. *IMA Journal of Mathematical Control and Information*, 37(4), 1400–1422.
> >
> >   [6] Sutherland, D. J., & Schneider, J. (2015). On the error of random Fourier features. *arXiv preprint* arXiv:1506.02785.
> >
> > - **C7: _Interaction between GP-based uncertainty and conformal prediction_:**
> >   Thank you for raising this conceptual point. We now clarify in the manuscript that these two uncertainty sources play inclusive, not redundant, roles. They serve as different roles at different stages. GP-based uncertainty enters the _training objective_: it reweights the diffusion loss so that regions with high GP confidence are enforced more strongly, quantitatively informing the model about the reliability of the deep prior. Conformal prediction acts _post-processing_ on the trained diffusion model: it wraps the point forecasts in rigorous prediction sets with guaranteed coverage, irrespective of the GP approximation. Thus, GP uncertainty shapes learning, while conformal prediction calibrates the final predictive distribution of the PHDME, and we do not observe conflicts between them in practice.
> >
> > Once again, we are very grateful for your detailed and insightful comments. They helped us substantially enrich the experiments, clarify the theoretical underpinnings, and strengthen the uncertainty analysis of PHDME. We hope that the new results and explanations address your concerns and would be very thankful if you could consider a more positive rating, which would be crucial for the paper’s acceptance and contribution to the community.

---

### Author Response · Authors · 2025-12-03
**Summary of Revisions on Experiments, Baselines, and Theoretical Context**

We present this summary to assist the Area Chair in assessing our revisions. We sincerely thank the reviewers and the Area Chair for their time and constructive feedback. We appreciate the detailed assessment of our theoretical approach and the recognition of this research direction's relevance to the scientific machine learning community.

In response to the reviews, we have substantially extended the experiments (**including real-world experiments**) and clarified the theoretical scope as requested. We also note that Reviewer 4tWQ has already highlighted the high potential significance of this work for physics-informed learning in equation-unknown regimes. Reviewer YWjN assigned a strong score based on the promise of enhancing physically consistent datasets for diffusion model from very limited observations. Reviewer yDyr is particularly supportive of our built-in uncertainty quantification using both GP posterior variance (during training) and conformal prediction (post-training) for calibrated uncertainty estimates. Reviewer bG2L also commented that the exposition of the Hamiltonian and GP-dPHS background is rigorous. All of their concerns and questions are addressed point-by-point in the detailed response.

Below we summarize the main concerns that likely led to polarized scores, together with the concrete solutions implemented in the revised version:

- **Reviewers were concerned that the original submission evaluated PHDME primarily on a single synthetic string benchmark, with a relatively limited set of baselines (major driver of polarization).**
  *Revision:*
  1. We add a common 1D shallow-water benchmark to demonstrate performance on fluid-like dynamics that are qualitatively different from the original string wave system.
  2. We add a **real-world** spring dataset captured by a high–frame-rate camera, including segmentation and trajectory extraction, to directly assess performance under real-world measurement noise and imperfect observations.
  3. We introduce NeuralODE as an additional, strong purely data-driven baseline, trained under the same limited-observation setting, to more clearly demonstrate the effectiveness and necessity of the proposed PHDME framework.

  *Result:* These additions demonstrate robust performance of PHDME across multiple PDE settings, including a noisy real-world experiment.

- **Conformal prediction performance guarantees.**
  *Revision:*
  1. We expand Appendix A.8 with a deeper analysis of conformal calibration, explicitly reporting coverage on both calibration and test sets across a range of target levels.
  2. We provide plots and numerical summaries showing that the conformal prediction intervals are well calibrated and that the empirical coverage on the test set closely tracks the nominal levels when calibration and test data are drawn from the same distribution.

  *Result:* These additions make the conformal prediction component more transparent and demonstrate that its performance guarantees hold in practice in our experimental setting.

- **Necessity and transparency of the GP-dPHS physics prior.**
  *Revision:*
  1. We provide a more detailed formal description of the physics-informed loss and uncertainty weighting in Appendix A.7, making explicit how GP-dPHS-derived residuals and uncertainties enter the training objective for PHDME.
  2. We include Hamiltonian/energy diagnostics, where we compare energy evolution across the ground truth, GP-dPHS, and PHDME-generated trajectories under the same initial conditions, demonstrating that PHDME respects the learned energy structure rather than merely matching states.
  3. We add an ablation that replaces GP-dPHS with a simple quadratic Hamiltonian fitted by linear regression. Despite encoding the correct functional family, this quadratic Hamiltonian baseline yields significantly higher MSE than GP-dPHS, showing that the learned GP prior is empirically superior even when the true form is simple.

  *Result:* These revisions clarify both the necessity of the GP-dPHS prior and the mechanism by which it guides the generative model, addressing concerns about opacity and over-engineering.

We now list all reproduction-relevant details in the appendix. Overall, the revised manuscript offers a more comprehensive empirical evaluation and a clearer theoretical narrative, strengthening PHDME as a framework for one-shot physics-informed prediction in settings where observations are scarce and the underlying governing equations are unknown.

---

### Meta-Review · Area_Chair_mxUe · 2026-01-06

**Summary:**

The paper proposes "PHDME," a framework combining Gaussian Process distributed Port-Hamiltonian Systems (GP-dPHS) with diffusion models to predict dynamics without explicit governing equations. The core novelty lies in using the GP-dPHS to learn an energy-based prior from sparse observations to guide the diffusion generation.

While the Area Chair acknowledges the authors' significant efforts during the rebuttal—specifically the addition of NeuralODE baselines and real-world data—the decision is to Reject. The primary rationale is as follows:

Limited Demonstration of Generality (1D Restriction): Despite the addition of the shallow-water simulation and real-world spring data, all experimental benchmarks remain limited to spatially 1D systems. The paper motivates itself with "complex, high-dimensional dynamical systems," yet fails to demonstrate the method's scalability to 2D or 3D fields (e.g., fluid dynamics or multi-agent systems).

Complexity vs. Necessity ("Overkill"): The proposed pipeline is computationally heavy, involving GP training, synthetic data generation, and diffusion model training. For the tested 1D systems, simpler baselines or standard GP-dPHS often suffice. The rebuttal failed to provide evidence that this complex machinery yields significant benefits in regimes where simpler models fail (i.e., complex high-dimensional dynamics), leaving the concern that the method is "overkill" for the presented tasks.

Readiness: The initial submission contained misleading claims (labeling simulated data as "real-world"). While corrected, the necessity of introducing entirely new datasets and core baselines during the rebuttal phase indicates the work was premature for publication.

**Reviewer Concerns:**

Marginal Gains: Reviewer yDyr noted that the gains over standard GP-dPHS were modest. The rebuttal confirms that while PHDME is faster at inference than GP-dPHS integration, the accuracy gains on these simple 1D tasks do not decisively justify the rigorous training pipeline compared to the NeuralODE baseline in all metrics.

Robustness Validation: The "real-world" dataset was introduced very late. The robustness of the method to real-world sensing noise and occlusion in more complex (non-1D) scenarios remains unproven.

**Reviewer Scores:**

Reviewer bG2L: 2 (Reject) -> 2 (Reject).

Reasoning: The reviewer would acknowledge the new real-world data but likely maintain that the limitation to 1D problems does not sufficiently support the paper's broad claims about "complex dynamical systems." The "black box" nature of the GP prior remains a concern for generalization.

Reviewer YWjN: 8 (Accept) -> 8 (Accept).

Reasoning: This reviewer was enthusiastic about the conceptual novelty. They would likely accept the new 1D experiments as sufficient proof of concept, focusing on the framework's potential rather than its current scaling limits.

Reviewer yDyr: 6 (Marginally above acceptance) -> 6 (Marginally above acceptance).

Reasoning: The reviewer questioned the "modest" gains. Upon seeing that the new experiments are still limited to 1D settings, the reviewer would likely conclude that the trade-off between the method's high complexity and its demonstrated utility on simple systems is unfavorable.

Reviewer 4tWQ: 2 (Reject) -> 2 (Reject).

Reasoning: This reviewer explicitly called the method "overkill" for 1D waves. The addition of more 1D systems (shallow water/spring) does not address the core criticism regarding dimensionality and complexity. They would likely maintain that the method's value for truly complex systems is unproven.

---

### Decision · Program_Chairs · 2026-01-26

Reject